# WARC-Bench: Web Archive Based Benchmark for GUI Subtask Executions

**Sanjari Srivastava, Gang Li, Cheng Chang, Rishu Garg, Manpreet Kaur, Charlene Y. Lee,**[*]
**Yuezhang Li, Yining Mao, Ignacio Cases, Yanan Xie & Peng Qi**
Uniphore
{sansri264@gmail.com,leemagpie@gmail.com,cheng.chang@uniphore.com}

## Abstract

Training web agents to navigate complex, real-world websites requires them to master *subtasks*—short-horizon interactions on multiple UI components (e.g., choosing the correct date in a date picker, or scrolling in a container to extract information). We introduce WARC-Bench (Web Archive Benchmark), a novel web navigation benchmark featuring 438 tasks designed to evaluate multimodal AI agents on subtasks. WARC-Bench enables sandboxed interactions with dynamic and realistic webpages using Web ARChive files. We show that WARC-Bench is challenging for leading computer-use models, with the highest observed success rate being 64.8%. To improve open source models on subtask, we explore two common training techniques: supervised fine-tuning (SFT) and reinforcement learning with verifiable rewards (RLVR). Experiments show that SFT models obtain a 48.8% success rate on the benchmark. Training with RLVR over SFT checkpoints, even in data-scarce settings, improves the score to 52.8% on WARC-Bench, outperforming many frontier models. Our analysis concludes that mastering these subtasks is essential for robust web planning and navigation, and is a capability not extensively evaluated by existing benchmarks. More details about WARC-Bench can be found at https://sanjari-orb.github.io/warc-bench/.

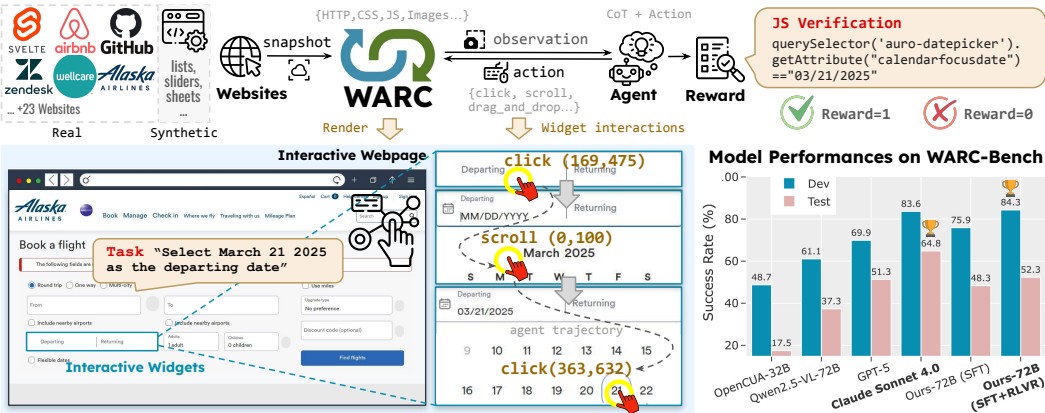

Figure 1: Overview of WARC-Bench. We record real and synthetic websites as Web Archive files to create interactive web environments for evaluating subtask execution and widget interactions in GUI Agents. WARC-Bench uses programmatic reward functions for automatic evaluation. Subtasks remain challenging for frontier models. Our model trained via SFT and RLVR achieves state-of-the-art results among open-source models.

## 1 Introduction

Autonomous agent systems that can execute complex web-based tasks (Gao et al., 2024; Ning et al., 2025) have attracted a lot of research interests and investment recently. Agentic web navigation

---

[*]Work done during internship

presents significant challenges, which researchers have approached from multiple angles (Gao et al., 2024). At one end of the spectrum, single-step visual grounding focuses on the ability to map textual descriptions to precise pixel coordinates within images (*e.g.* "*Output the pixel coordinates of the button 'Japanese' under language options*") (Cheng et al., 2024; Liu et al., 2024a). On the other end, comprehensive web navigation benchmarks such as WebArena (Zhou et al., 2023) and Mind2Web (Deng et al., 2023a) are created to assess the performance of web agents in solving complex multi-step and long-horizon browser-based workflows (*e.g.* "*Place an order for a pair of green women's Levi's jeans under $50 on Amazon.*").

Complex browser-based tasks typically involve constituent steps, like exploring the webpage via scrolling, interacting with UI widgets (dropdowns, date pickers, menu bars, etc.), extracting information, or recovering from navigational errors. To our knowledge, this intermediate level of agentic complexity remains unaddressed in existing realistic GUI benchmarks. In this work, we focus on investigating agent performance on these intermediate **subtasks** — *i.e.*, **smaller workflow components that correspond to simple natural language instructions** within a larger web browsing task. Subtasks typically involve multiple primitive UI actions, such as click, scroll, type, and mouse move. Some examples include "*Select 10/10/2025 as the start date*" or "*Reply to the comment with a 'Thank you' in bold font*".

To this end, we develop **WARC-Bench** (**W**eb **ARC**hive Benchmark), a novel benchmark designed to evaluate the accuracy of multimodal AI agents in executing subtasks on web browsers. We provide realistic web environments that fully replicate real-world websites, and serve them for Web agents to interact with and complete tasks. To achieve this, WARC-Bench uses Web Archive (WARC) files to record and replay websites in a Chromium-based browser, allowing web agents to interact with the webpage flexibly. Our benchmark dataset consists of (a) WARC based **web environments**, (b) per-task **subtask goal** in natural language, and (c) per-task programmatic (thus, deterministic) **reward function**. The verifiable reward functions are used to measure task completion at the end of the trajectory. This makes the evaluation independent of the particular path an agent takes to complete the task. WARC-Bench contains a manually labeled test set of 200 real-world examples, and a train/development set of 1059/238 examples consisting of synthetic and real-world subtasks.

We use WARC-Bench to evaluate leading large vision-language models (VLMs) and computer-use agents on subtask execution in realistic websites. We find that WARC-Bench poses a significant challenge to current large foundation models. Specifically, Anthropic's Claude-4.0-Sonnet model[1] achieves the highest performance, with **64.8%** task completion on the test set, followed by OpenAI's GPT-5[2] at 51.3%. Open-source models lag considerably behind, with Qwen-2.5VL-72B (Bai et al., 2025) achieving the best performance among them at 37.3%.

To improve open source models, we construct synthetic datasets to fine-tune Qwen2.5-VL (Bai et al., 2025) family of models on subtask execution. Our 72B model trained with Supervised Fine-Tuning (SFT) and Reinforcement Learning with Verifiable Rewards (RLVR) achieves **52.3%** accuracy on the test set, significantly outperforming other open-source models and most frontier closed-source models. We also conduct comparative analysis of the models on other GUI navigation benchmarks like WebArena (Zhou et al., 2023), Screenspot V2 (Wu et al., 2024) and MiniWoB++ (Liu et al., 2018a), highlighting the unique contributions of the WARC-Bench to GUI agents research.

To summarize, the key contributions of this paper are four-fold:

- We formally define GUI **subtasks** - short-horizon tasks that constitute broader browser-based workflows. We create **WARC-Bench**, a novel, lightweight, extensible benchmark that, to the best of our knowledge, is the first Web-Archive file based benchmark designed to measure realistic GUI subtask execution accuracy.
- We find that WARC-Bench poses a challenge to state-of-the-art large foundation models, with Anthropic's Claude-Sonnet 4.0 achieving the highest success rate of 64.8%.
- We show that high-fidelity subtask performance on WARC-Bench better tracks long-horizon web navigation capabilities when compared to other benchmarks like Screenspot-V2 and MiniWoB++, demonstrating that WARC-Bench fills a gap in the existing landscape of web benchmarks.

---

[1]https://www.anthropic.com/news/claude-4
[2]https://openai.com/index/introducing-gpt-5

| Dataset | #Tasks (Templates) | Interactive Envs | Task Isolation | Deterministc End State Rewards | Diverse Observation Space | Scalable Design |
|---|---|---|---|---|---|---|
| Multimodal-Mind2Web | 1013 (—) | ✗ | ✓ | ✗ | ✗ | ✗ |
| Online-Mind2Web | 300 (136) | ✓ | ✗ | ✗ | ✓ | ✗ |
| WebArena | 812 (241) | ✓ | ✗ | ✓ | ✓ | ✗ |
| OSWorld | 369 (—) | ✓ | ✓ | ✓ | ✓ | ✗ |
| **WARC-Bench** | 438 (—) | ✓ | ✓ | ✓ | ✓ | ✓ |

Table 1: Comparison of Multimodal Benchmarks for Web Task Performance of GUI Agents. (—) templates indicate that all benchmark tasks are unique.

- We explore SFT and RLVR to train competitive 7B/72B parameter models that outperform open-source and many leading closed-source computer-use models on subtask execution. Specifically, we demonstrate that RLVR with synthetic subtasks can improve agent performance on real-world subtasks, with enhanced visual grounding capabilities, better exploration, and contextual awareness.

## 2 WARC-BENCH: WEB ARCHIVE BASED ENVIRONMENTS FOR TESTING SUBTASK EXECUTION BY GUI AGENTS

Our work seeks to address a gap between visual grounding and complex web navigation - specifically, short-horizon tasks that consist of more than one primitive UI action that appear as a functionally singular instruction to humans. We call such tasks **GUI subtasks**. They satisfy the following constraints.

- They achieve a self-contained and unambiguous goal using browser-based actions;
- They can be completed by a human within 1-20 atomic UI actions (Table 6).

GUI subtasks require skills such as navigating menu options, setting dates on datepickers, scrolling a webpage to perform entity extraction, filling form fields, editing spreadsheet cells, and setting values on drag-and-drop slider bars, among many others. Compositional frameworks such as Agent-S2 ((Agashe et al., 2025)) achieve strong performance on complex tasks by breaking them down into subtasks. This provides evidence that specialist models with high subtask accuracy can yield substantial improvements in general, long-horizon GUI agents as well. Our objective is to systematically examine the capability of leading large VLMs in completing such subtasks by constructing a new benchmark called WARC-Bench.

**WARC-Bench** is an extensible GUI agent evaluation suite containing realistic, interactive, and lightweight web environments that can be used to perform a wide spectrum of short-horizon tasks. We require that our web environments capture real-life, complex UI widgets and densely populated webpages. To do so, we rely on **W**eb **Arc**hive (WARC[3]) files to render interactive websites within our evaluation suite. Web Archives are snapshots that can preserve the complete state of the website during the period of recording. They enable high-fidelity replay of archived webpages by recording assets like HTML, CSS and JavaScript (JS) files, embedded contents like images/videos and even HTTP headers and metadata for replicating network calls. The recorded trajectories are fully replayable, resulting in an realistic and interactive clone of the webpage. This is uniquely well-suited to evaluating and training agents on short-horizon tasks. Replaying WARC files instead of live websites also makes the benchmark easy and fast to run.

Using web archives as benchmark environments has the following advantages (Table 1):
**High fidelity with real websites** - WARCs allow us to realistically replicate websites.
**Task isolation** - Each task run uses a unique copy of an archived benchmark environment. This ensures proper isolation and sandboxing for each task. Using WARC files also prevents us from making any real write operations on actual websites while running the benchmark.
**Scalable design** - The benchmark is designed to be extensible and easy to evaluate. Unlike prior works that are constrained by simulated environments (*e.g.*WebArena, OSWorld), adding a new environment to WARC-Bench simply means adding a new web recording to the test suite.

---

[3]https://en.wikipedia.org/wiki/WARC_(file_format)

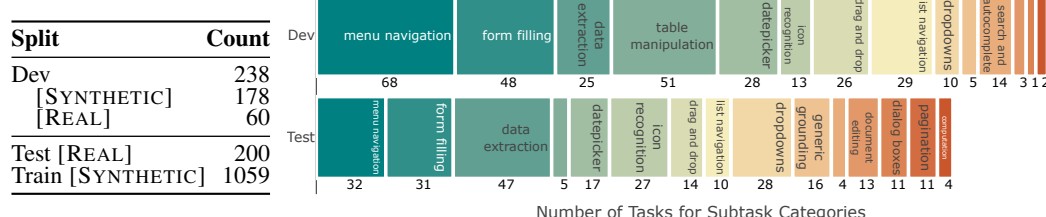

| Split | Count |
|---|---|
| Dev | 238 |
| [SYNTHETIC] | 178 |
| [REAL] | 60 |
| Test [REAL] | 200 |
| Train [SYNTHETIC] | 1059 |

Figure 2: Statistics and Distribution of Subtasks in WARC-Bench. Each task in WARC-Bench can belong to multiple subtask categories, here we illustrate the category coverage in the figure on the right.

We develop a lightweight WARC replayer that can simulate the archived website in a Chromium-based automated browser created using Playwright.[4] This is conceptually similar to existing tools like ReplayWeb.page[5] that allow replaying web archives in the browser. WARC-Bench tasks are runnable as Gym (Brockman et al., 2016) environments, facilitating easy integration into existing evaluation and training frameworks like BrowserGym (de Chezelles et al., 2025) and Volcano-Engine Reinforcement Learning library for Agents (*verl-agent* by Feng et al. (2025)).

**Limitations of WARC-based environments:** Any webpage interactions that were not recorded or cannot be rendered from the stored HTML/JS contents are not replayable. If all assets required to access the target URI are found, the webpage can be rendered exactly the same as the original page, otherwise our replay server displays an error. Websites that use Cloudflare or block robots are also often not archivable. The HTML/JS contents are sometimes indexed against URIs containing fields like timestamps, random numbers or session IDs that require careful handling during replays.

## 2.1 TASK BREAKDOWN

In this section, we dive into the details of the tasks present in WARC-Bench and the data collection strategies we used to assemble this benchmark. Each task within WARC-Bench dataset consists of the following components: (1) **Environment**: A realistic web environment in which the agent operates. Each environment is backed by a single WARC file, with additional specifications including the starting URL, and optionally the timestamp of when the WARC file was captured. (2) **Goal**: An *unambiguous, natural language* description of the subtask that the agent must perform within the web environment. Given the goal, the agent should either explore the current webpage to determine a correct next action or choose when to terminate the trajectory. The goal should result in a deterministic and unique end-state of the webpage. Some example subtask goals are, '*Enter "MacBook Pro" in cell A1*' or '*Select 02/03/2025 (mm/dd/yy format) as the start date*'. (3) **Evaluator**: Code-based evaluation criteria that determines the trajectory level reward. Each evaluator is defined by the *evaluator type* and an *evaluator function*, enabling deterministic measure of the task success. We support 4 evaluator types - (a) JS function-based evaluator, (b) URL matcher, (c) String matcher , and (d) JSON matcher. The latter two evaluators are primarily used for information seeking tasks. Examples include, `document.querySelector('#riskslider').value=='4'` (JS Evaluator) or `{'total_tow_fee': 657}` (JSON Matcher).

## 2.2 DATA COLLECTION METHODS

To determine which GUI subtasks to include in the benchmark, we compile a list of 15 common subtask categories that require multi-step interactions within webpages (see Figure 2). We then select 29 real-world websites that are recordable as web archives and have a good representation of our task categories. The websites we selected include *Github*, *Zoho Desk*, *Zendesk*, *Kaiser Permanente*, *Google Earth*, *Scrimba*, etc. The goals and evaluator functions for real websites are collected by the authors manually.

We also create 62 synthetic websites through an LLM-based pipeline that can create HTML pages with a diverse representation of interesting UI widgets. This approach has also been explored in

---

[4]https://github.com/microsoft/playwright
[5]https://github.com/webrecorder/replayweb.page

concurrent works like FormFactory (Li et al., 2025). The LLM-based pipeline lets us automatically and cheaply generate environments, task goals, and evaluator functions for the benchmark. The goals and evaluator functions generated in this manner still need to be manually vetted for feasibility and correctness, but the process is a lot faster than end-to-end manual data labeling.

We create a total of 1497 subtasks (1059 training, 238 development, 200 test) on the collected environments for WARC-Bench. We constrain the test dataset (200 tasks) to only use real web environments, while the development set contains a mix of 60 real and 178 synthetic tasks. We also generate an illustrative, synthetic training set for online reinforcement learning. We observe that agent performance is lower on real tasks when compared to synthetically generated tasks, underscoring the need for including manually collected, high quality test websites in web navigation benchmarks (Table 2).

## 3 WEB AGENTS FOR SUBTASKS

In this section, we formalize the concept of AI Agents for computer-use and web navigation (web agents) and introduce Subtask Vision Agent (SVA) – a generalist VLM based web agent we created for subtasks. We also go over leading computer-use models that we evaluate with WARC-Bench.

### 3.1 FORMULATION

Given a website (*e.g.* www.dmv.ca.gov) and a goal (*e.g.* *"Set the language of the page to Portugese"*), we define a web agent to be a software program that can autonomously generate a sequence of actions and complete the goal on a graphical user interface (*e.g.* web browser on desktop/mobile). Since a website typically transforms once an atomic action (Table 6) is performed on it, a web agent usually observes the state of environment, and greedily predicts the best atomic action that brings it closer to achieving the goal (Qin et al., 2025; Yao et al., 2023). It also terminates the trajectory once the task is complete or is deemed infeasible. Thus, we can formalize the task completion problem by web agents as: at each step of the trajectory, given an initial goal, an observation of the current environment, an action space, and a history of previous observations and model decisions, find the optimal next action that would eventually result in the success of the task.

### 3.2 SUBTASK VISION AGENT (SVA)

In order to evaluate and compare the proficiency of vision-language models [6] at subtask execution, we design a simple yet powerful web-navigation agent called Subtask Vision Agent (SVA) (Figure 3). At each step of the subtask trajectory, SVA takes in the *goal*, current *screenshot* of the webpage, the *action space*, and a *history* of previous steps. It then produces a Chain-of-Thought (CoT) and the next action within the response. The complete system and user prompt of the agent and CoT questions can be found in Appendices F and G.

SVA uses only screenshots to observe the state of the webpage. This makes textual representation of the webpage such as accessibility trees and HTML DOM redundant. Furthermore, text observations are lengthy and can exceed the context limit of certain models. Modern HTML websites use complex visual elements like icons, widgets, and canvas-based rendering, making screenshots a more faithful representation of the interface than text. The approach of using vision-only observations aligns with the methodologies employed by all major vision-language agents mentioned in Section 3.3.

The history provided to the agent has the screenshots and the model responses at previous steps of the trajectory (more details in Appendix F). We limit the length of the history observed by the backbone model to 5.

We include a minimal API action space in the system prompt with eight (8) types of actions: *click, complete, drag_and_release, hover, key_press, scroll, type, wait*. The action space descriptions provided to the agent can be found in Appendix G. We use BrowserGym (de Chezelles et al., 2025)[7] as the underlying engine to interact with the browser.

---

[6] We do not focus on text-only Large Language Models in this work.
[7] https://github.com/ServiceNow/BrowserGym

Figure 3: Diagram of the Subtask Vision Agent (SVA) design.

We find that SVA is an effective agent design with high task completion performance despite using significantly lower tokens and fewer trajectory steps on average as opposed to other computer-use agents (Table 2, Figure 5).

### 3.3 EXISTING AGENT DESIGNS

For baseline experiments, we evaluate 4 proprietary computer-use agents – Anthropic's Computer-Use agent (Claude CUA)[8], OpenAI's Computer-Using Agent (OpenAI CUA)[9], Bytedance's open source computer-use model UI-TARS 1.5 7B (Qin et al., 2025) and the open-source OpenCUA computer-use models by Wang et al. (2025). All these computer-use models contain their respective, pre-defined action spaces. All four agent designs follow an *observe-predict-act* loop, *i.e.*, fetching observations from the web environment followed by selecting the next best action until task completion. The actuation of predicted action for all CUA agents is implemented by us using BrowserGym. We provide the prompt used in the agentic evaluation of these models in Appendices H and I.

## 4 EXPERIMENTS

To find whether subtask execution is challenging for foundational models, we perform extensive evaluations using the development and test splits of WARC-Bench. We evaluate open-source, proprietary, and computer-use models from ByteDance, OpenAI, and Anthropic with their recommended agent design as well as with the Subtask Vision Agent (SVA). Through our analysis, we show that SVA is a competitive agent design for subtasks (Table 2, Figure 5).

We discover that GUI subtasks are especially challenging for smaller and medium-scale open-source VLMs. To address this, we train Qwen2.5VL family of models by Bai et al. (2025) (7B and 72B parameters) with curated training datasets to enhance subtask accuracy. All our models are trained with the SVA prompt architecture. All the screenshots used in the web state observation and the past history, are of the resolution 1280x720 pixels each.

**Distillation from Teacher Model Trajectories**: We created a large scale SFT dataset using a novel and scalable subtask trajectory crawling framework. This allows us to collect around 12k unique trajectories using teacher models with strong UI interaction capabilities. These trajectories are used to distill grounding, planning, error correction, and reward modeling capabilities into the Qwen2.5-VL models. The sources for websites used for trajectory collection include – Common Crawl [10], webpages for UI component libraries, and synthetically created websites.

**Reinforcement Learning with Verifiable Rewards**: As described in Section 2.2, we develop a LLM based data synthesis pipeline to automatically create WARC-Bench tasks – including synthetic webpages, subtask goals and evaluation rules. This approach is used to create an illustrative training dataset of Gym-environment based subtasks (1059 tasks). To improve on the supervised fine-tuned model, we further apply Proximal Policy Optimization (PPO) algorithm (Schulman et al., 2017) to learn from the agent rollouts with environment feedback directly. At each training step, the agent generates rollouts on a batch of tasks. We use a simple reward scheme where the agent gets a positive reward 10 for successful trajectories and 0 for failed or truncated trajectories due to exceeding max steps. The list of hyperparameters can be found in Appendix D.

---

[8]https://docs.anthropic.com/en/docs/agents-and-tools/tool-use/computer-use-tool
[9]https://platform.openai.com/docs/guides/tools-computer-use
[10]https://commoncrawl.org/

| Model | Accuracy (%) | | | |
|---|---|---|---|---|
| | Dev[SYNTHETIC] | Dev[REAL] | Dev[TOTAL] | Test |
| OpenAI computer-use-preview (*2025-03-11*)[CUA] | 62.17 | 49.44 | 58.96 ±1.35 | 33.83 ±2.58 |
| GPT-4o (*2024-11-20*) | 7.87 | 14.51 | 9.54 ±1.70 | 9.17 ±1.04 |
| GPT-5 (*2025-08-07*) | 72.66 | 61.67 | 69.89 ±1.21 | 51.33 ±3.23 |
| Claude Sonnet 4.0 (*2025-05-14*)[CUA] | 79.92 | 76.11 | 78.96 ±1.11 | 47.17 ±2.89 |
| Claude Sonnet 3.7 (*2025-02-19*) | 82.96 | 78.89 | 81.93 ±1.51 | 59.83 ±1.26 |
| Claude Sonnet 4.0 (*2025-05-14*) | 84.27 | **81.67** | 83.61 ±0.84 | **64.83** ±1.61 |
| Qwen2.5-VL 7B | 16.85 | 11.67 | 15.54 ±0.42 | 4.67 ±1.04 |
| UI-Tars 1.5 7B[CUA] | 44.01 | 26.55 | 39.66 ±0.42 | 10.33 ±0.76 |
| OpenCUA 7B[CUA*] | 48.03 | 41.67 | 46.43 | 14.00 |
| **Ours-7B-SFT** | 70.60 | 54.49 | 66.54 ±0.57 | 27.33 ±1.47 |
| **Ours-7B-RLVR** (SFT+RLVR) | 78.09 | 54.44 | 72.13 ±0.64 | 29.17 ±1.15 |
| OpenCUA 32B[CUA*] | 51.12 | 41.67 | 48.74 | 17.50 |
| Qwen2.5-VL 72B | 64.23 | 51.67 | 61.06 ±1.99 | 37.33 ±0.85 |
| **Ours-72B-SFT** | 78.23 | 68.89 | 75.88 ±0.97 | 48.33 ±1.66 |
| **Ours-72B-RLVR** (SFT+RLVR) | **87.64** | 76.67 | **84.31** ±0.87 | 52.33 ±0.76 |

Table 2: Trajectory-Level Success Rates on WARC-Bench. Small VLMs (7B params) are in gray. Results are divided into closed (top) vs. open-source (bottom) models. [CUA] indicates models evaluated with the provider's computer-use agent. All other models use SVA. All values are averages across 3 runs, however [*] indicates single-run results due to prohibitive HuggingFace inference costs for OpenCUA. Best per benchmark is in **bold**. Best inside its sector (small/large open source vs closed source) is underlined.

## 5 RESULTS AND ANALYSIS

Table 2 shows the main results of various model-agent combinations evaluated on WARC-Bench. Our main findings are: **(1) WARC-Bench is a challenging benchmark for leading frontier models.** While Claude-based agents show superior performance to other computer-use agents and models, the best Claude model only achieves 64.83% success rate on the benchmark, leaving significant room for future improvement. **(2) SVA is a simple yet competitive agent design.** With the same family of frontier models, GUI agents based on the SVA design significantly outperform their dedicated computer-use agent counterparts, including proprietary ones. **(3) Distilling from strong frontier models can effectively boost open-source model performance on WARC-Bench.** With SFT training, we see significant improvements over the original Qwen2.5VL 7B and 72B models from 4.67% to 27.33% (Ours-7B-SFT) and 37.33% to 48.33% (Ours-72B-SFT). **(4) Agentic RLVR training significantly improves agent performance further beyond SFT models.** With additional RLVR training on the synthetic training set, we see further performance boost over the SFT baselines for both the 7B (27.33% to 29.17% Ours-7B-RLVR) and 72B (48.33% to 52.33% Ours-72B-RLVR). Further analysis on the development set shows that training on synthetic data not only boosts success rate on synthetic tasks (*+9.41%*), but also their real-website counterparts (*+7.78%*).

**Effects of Agentic RLVR.** We analyze the behavioral patterns of our 72B SFT and RLVR models on the WARC-Bench development split. A breakdown by task categories (Figure 4a) reveals that the RLVR model achieves substantial improvements in dynamic tasks such as *form filling*, *menu navigation*, *table manipulation*, and *datepicker*, while maintaining comparable or slightly higher performance in most other categories. We observe that these gains are driven by enhanced vision grounding capabilities and better exploration / contextual awareness. Specifically, the RLVR model demonstrates greater precision in identifying small interface elements (e.g., calendar dates, nested menu items) and reduced errors from misplaced clicks. Agent action distribution (Figure 4b) shows increased use of scrolling actions for better exploration of the webpage by the RLVR agent, and also reveals a clear efficiency gap between the two agents: the RLVR model executes fewer overall actions, reduces redundant clicks, and leverages compound operations (e.g., click-and-type) more consistently. As a result, the RLVR model produces shorter trajectories on average, with approximately 0.94 fewer step per task compared to the SFT model (Figure 4c).

**Comparison to other GUI navigation benchmarks.** We also compare agent performance on several other web-navigation benchmarks to compare with WARC-Bench, including long-horizon tasks

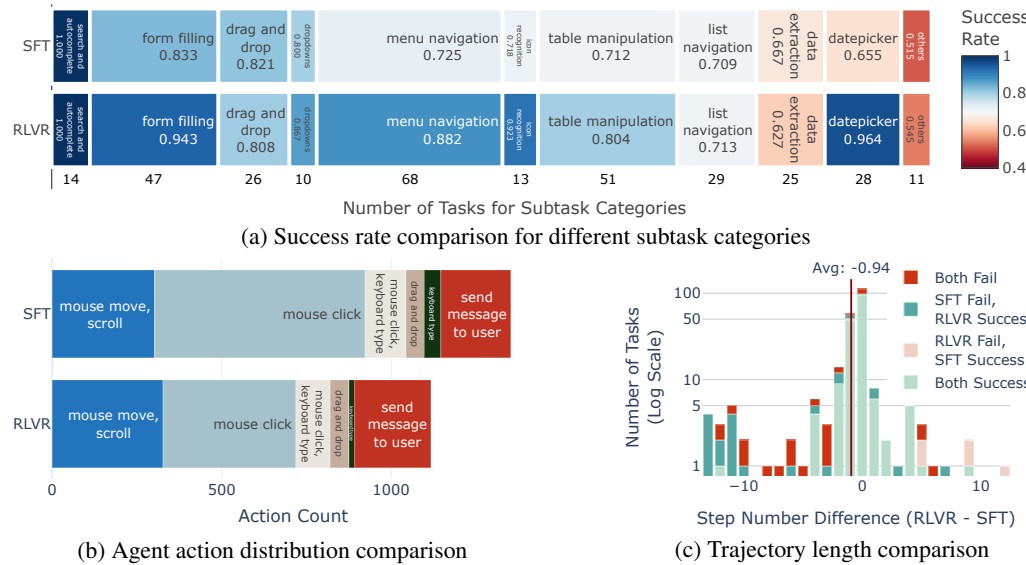

(a) Success rate comparison for different subtask categories

(b) Agent action distribution comparison

(c) Trajectory length comparison

Figure 4: Behavioral analysis of Ours-72B-SFT (SFT) v/s Ours-72B-RLVR (RLVR) model

(WebArena; Zhou et al., 2023), synthetic, low-fidelity widget tasks (MiniWoB++; Liu et al., 2018a), and GUI grounding tasks (ScreenSpot V2; Wu et al., 2024; Cheng et al., 2024).[11]

| Model | WARC-Bench (test) | WebArena (no map) | Miniwob++ | ScreenSpot V2 |
|---|---|---|---|---|
| Qwen 2.5 VL 7B | 4.67% | 3.07% | 12.53% | 51.62% |
| Qwen 2.5 VL 72B | 37.33% | 15.68% | 53.87% | **88.05%** |
| GPT-5 (*2025-08-07*) | 51.33% | 34.06% | 52.27% | 26.39% |
| Claude 4 Sonnet | **64.83%** | **37.96%** | **71.73%** | 85.06% |
| Ours-7B-RLVR | 29.17% | 7.31% | 36.27% | 75.81% |
| Ours-72B-RLVR | 52.33% | 26.80% | 59.20% | 82.44% |

Table 3: SVA agent performance comparison across benchmarks. Each number is an average of 3 runs. Best per benchmark is in **bold**. Second places are underlined.

Table 3 shows the results of task performance comparison across benchmarks. Our main findings are: (1) **Agent performance on grounding and low-fidelity widget tasks correlate poorly with long-horizon task performance, while WARC-Bench tracks it at a system level.** This can be seen by comparing the relative ranking of agents by the performance on each benchmark, as well as relative performance gaps. Most notably, while vanilla Qwen 2.5 VL models achieve stronger performance than frontier models, they are nevertheless least effective at long-horizon tasks. (2) **Models fine-tuned over subtask data improve over their base models on all tasks (except for grounding).** As can be seen, fine-tuning models on subtask data can improve agent performance on both low-fidelity widget tasks and long-horizon web navigation tasks. While the former is more expected due to the similarities in task setup, the gains on long-horizon tasks shows that training with subtask objective improves a model's overall web navigation capabilities. Together with the observation that WARC-Bench performance tracks that of WebArena, this finding corroborates our hypothesis that solving subtasks well is a precursor to agent improvement on longer-horizon web navigation tasks. (3) **WARC-Bench provides stress testing on a crucial GUI agent capability that is not well-covered by existing benchmarks.** As shown in Table 3, while GPT-5 and Claude 4 Sonnet agents achieve comparable performance on WebArena (within about 11% relative performance gap), their performance gap on WARC-Bench is enlarged to about 26%, indicating key

---

[11]We omit the OpenStreetMap portion of the WebArena benchmark due to technical difficulties with the online map service. For Screenspot V2, which is a visual grounding benchmark, we limit the agent action space to `click(...)` only. For each evaluation, we use the same prompt format built inside our Subtask Vision Agent (see Appendix F).

generalization capabilities uniquely captured and assessed by WARC-Bench. For long-horizon tasks like WebArena to capture the same level of comprehensiveness and diversity, a great deal of efforts are necessary for task creation and environment maintenance. WARC-Bench sidesteps this limitation with lightweight, sandboxed WARC interactive environments, and ensures ease of scaling on task diversity and difficulty to stress-test GUI agents on this important aspect.

Overall, these observations demonstrate that WARC-Bench is a unique benchmark for GUI agent development, and one that helps advance agent performance on long-horizon tasks, which is closer to the ultimate goal of GUI agents.

**Decomposition of web navigation into planning and subtasks.** One motivation for this work is to show that GUI sub-agents that specialize in executing subtasks can bring gains to end-to-end GUI navigation tasks, making subtasks a critical capability to evaluate, diagnose, and improve upon in generalist web agents. A common approach to managing agent context for a long-horizon, end-to-end web agent is to create a hierarchical agent with a Planner (P) sub-agent (prompt provided in Appendix J) that can break down a long-horizon task into subtasks and redirect them to a performant subtask model (e.g., **Ours-72B-RLVR w/ SVA**) acting as an executor (E) sub-agent. Table 4 shows how choosing the correct agent design for web navigation tasks can bring accuracy gains, while offloading most of the token footprint to cheaper, open source models. We use a subset of 165 WebArena (Zhou et al., 2023) filtered for evaluator correctness called WebArena-Lite (WA-Lite) (Liu et al., 2024b) for these experiments.

- For Qwen2.5-VL-72B based agents, we can significantly improve the performance (+8.46%) by adopting a hierarchical planner+executor agent design and using a strong executor model (Ours-72B-RLVR) [Rows 1, 3].

- For strong models like Claude-4.0-Sonnet, breaking down the task into high level plans v/s subtask executions does not bring any gains/downsides in overall performance [Rows 4, 6]. However, even for Claude-4.0-Sonnet, we can still benefit from a hierarchical design by offloading most of the agent tasks to a capable executor model (Ours-72B-RLVR) with no drop in accuracy at a potentially lower cost of inference [Rows 4, 6, 7].

- For long-horizon tasks, the quality of the planner model plays a significant role in determining the success rate [Rows 2, 5 and Rows 3, 7]. Using a better planner like Claude instead of Qwen2.5-VL-72B, with the same capable executor (Ours-72B-RLVR), significantly improves WA-Lite SR by +11.19% points [Rows 3, 7].

| # | Planner Model | Subtask Executor | Agent | WA-Lite SR% |
|---|---|---|---|---|
| 1 | N/A | Qwen2.5-VL-72B-Instruct | SVA | $16.17 \pm 1.14$ |
| 2 | Qwen2.5-VL-72B-Instruct | Qwen2.5-VL-72B-Instruct | Hier. [P, E] | $22.39 \pm 2.58$ |
| 3 | Qwen2.5-VL-72B-Instruct | Ours-72B-RLVR | Hier. [P, E] | $\mathbf{24.63 \pm 0.75}$ |
| 4 | N/A | Claude-4.0-Sonnet | SVA | $\mathbf{36.32 \pm 1.14}$ |
| 5 | Claude-4.0-Sonnet | Qwen2.5-VL-72B-Instruct | Hier. [P, E] | $31.84 \pm 0.87$ |
| 6 | Claude-4.0-Sonnet | Claude-4.0-Sonnet | Hier. [P, E] | $35.32 \pm 1.14$ |
| 7 | Claude-4.0-Sonnet | Ours-72B-RLVR | Hier. [P, E] | $35.82 \pm 1.29$ |

Table 4: Web navigation accuracies on a simple SVA agent v/s Hierarchical Planner+Executor agent designs. SVA – Single-loop execution agent detailed in Section 3.2. Hier.[P,E] – Hierarchical agent consisting of a Planner that determines what subtasks to perform, and an SVA agent acting as an Executor. Executor completes the subtask before returning control to the Planner.

**Human Baseline.** We performed human evaluation on the 60 realistic tasks in the development set (Dev-Real) with 3 participants. The persons who performed the evaluation were not involved in the dataset creation. The human accuracy we obtained is $87.11\% \pm 1.84\%$ (52.3 / 60 tasks). Among the failed tasks, tasks requiring reading long text or formatted outputs are most challenging for humans. For example, they failed the towing fee extraction task where one needs to scan through rules / tables of fees to come up with the final number. For tasks that require formatted outputs, sometimes the answers were written in slightly wrong format.

## 6 RELATED WORK

**Language agents for GUI automation.** Large language models (LLMs) powered AI agents have shown to be capable of interacting with digital environments and completing various tasks on graphical user interfaces (GUIs) (Xie et al., 2024a). LLMs can understand web states (screenshots, HTML codes, etc.) and output code to execute actions on GUIs (Xie et al., 2024b; Sager et al., 2025). Latest open source works like Aguvis (Xu et al., 2025) and OpenCUA (Wang et al., 2025) have designed vision-only GUI agents that utilize VLM (Vision-Language Models) model's capabilities for planning and reasoning to correctly solve their tasks. Furthermore, due to the verifiable, sequential and exploratory nature of GUI tasks, many latest works such as GUI-R1 (Luo et al., 2025) and UI-R1 Lu et al. (2025) have applied Reinforcement Learning Post-Training techniques to their models.

**GUI agent benchmarks.** Existing GUI agent benchmarks span a spectrum of task complexity from single-step grounding tasks to long-horizon task completion in live or simulated environments. On the one end, visual grounding tasks (Cheng et al., 2024; Liu et al., 2024a) can be completed with a single primitive UI action (click, type, scroll) at specific pixel coordinates. While these evaluate agent capabilities to accurately identify the GUI coordinates for a single-step action, performance on these do not necessarily correlate with downstream task performance. On the other end, long-horizon tasks aim to test GUI agents on realistic tasks within practical websites or operating systems. WebArena (Zhou et al., 2023) employs five realistic, standalone and self-hostable web environments for building and evaluating autonomous agents that interact live with these websites. Multimodal-Mind2Web (Deng et al., 2023b; Zheng et al., 2024) features complex web task trajectories on real websites, but adopts an *offline* evaluation setting for per-step action prediction accuracy. Online-Mind2Web (Xue et al., 2025) re-applies Multimodal-Mind2Web tasks to a diverse set of real-world user tasks on real, online websites and uses an LLM-as-a-judge to evaluate task completion accuracy. On operating systems, OS-World (Xie et al., 2024b) is a popular benchmark containing real OS environments evaluating agent performance on open-ended computer tasks spanning multiple applications. These long-horizon tasks can better help evaluate how well GUI agents fulfill realistic user tasks on these environments, but are typically more difficult to develop on, and lack diversity and stress testing for their component subtasks. The closest family of previous GUI agent benchmarks to WARC-Bench are based on simplified/synthetically generated web environments focusing on UI widgets or simple tasks, such as MiniWob++ (Liu et al., 2018b) and FormFactory (Li et al., 2025). These tasks often lack the fidelity of real-world UIs, which can result in generalization gap when used to develop real-world GUI agents. WARC-Bench provides a high-fidelity, short-horizon, and interactive environment, which marks a unique contribution on this spectrum.

## 7 CONCLUSION

We present WARC-Bench, a novel and realistic benchmark for GUI subtasks. We demonstrate that subtask execution remains a challenging problem for frontier models like Claude 4.0 Sonnet and GPT-5. With synthetically generated training datasets, we show that open-source models trained with supervised fine-tuning (SFT) and agentic reinforcement learning with verifiable rewards (RLVR) achieve competitive performance with frontier models. WARC-Bench presents unique challenges and opportunities for agent development on GUI navigation, and complements existing long-horizon benchmarks with scalable, sandboxed environments.

We believe WARC-Bench opens up several promising research directions. For future work, we plan to better automate WARC-Bench sample creation and curation, study SVA performance as part of a larger agent framework for long-term tasks, and explore advanced training methods to further improve agent performance. We hope that our work will inform future studies of GUI subtasks and eventually lead to better generalized systems of computer-use.

## REPRODUCIBILITY STATEMENT

We use several open-source software packages in this study, including BrowserGym for web navigation (de Chezelles et al., 2025),[12] Volcano Engine Reinforcement Learning for LLMs (VERL) for SFT and offline RL training,[13] and verl-agent for agentic RL training.[14] Our large-scale data collection utilize the common crawl website index.[15]

The codebase for running WARC-Bench can be found at https://github.com/sanjari-orb/warc-bench. The codebase of training recipes for our SFT/PPO/GRPO 7B and 72B models can be found at https://github.com/sanjari-orb/orby-verl-agent. We separately discuss the hyperparameters of our SFT and RLVR training in Appendices C and D.

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

## A    SUBTASK ENVIRONMENT AND TASK EXAMPLES

We provide 10 examples of tasks based on real websites in the dataset Dev[MANUAL] below. The goals are made as unambiguous as possible to ensure validity of the evaluator functions.

> - Go to Button types in side menu and move slider to 40
> - Select date 6 June 2024 in the date picker
> - Select the font Times from the dropdown
> - Select March 21 2025 as the departing date and April 21 2025 as the returning date
> - Set the adults count to 3 and add 1 child of age 14
> - Continue filling the search form using the following information: "one way": True, "use miles": True, "upgrade": "MVP Gold 75K"
> - Swap the From and To airports in 1 step
> - Display only the tickets with "open" status. Keep the filter dropdown menu open for the user to verify your selection.
> - Go to the Organizations page
> - What were the total number of property crimes and property crime rate per 100000 population commited in year 2020? Give the answer in JSON format with the keys "total_crimes" and "crime_rate_per_100000". Do not include any commas in the values.

Similarly, below are 10 examples from Dev[SYNTHETIC].

> - Enter 'MacBook Pro' in cell A1
> - Select cell E1 in the spreadsheet and apply bold formatting using the toolbar
> - Navigate to Data Analytics tab and search for '2023' to find all employees who started in 2023
> - Type the formula '=C10*D10' in cell E10 to calculate total value
> - Sort the Employee Management Table by 'Status' column header in ascending order
> - Scroll down in the employee directory to see all employees
> - Drag the Marketing Budget slider to $75,000
> - In the Expense Reports tab, check the "Recurring Expense" checkbox in expense type options

- Fill in the employee registration form with first name 'Dr. Sarah' and last name 'Chen-Williams'
- Delete the last row from the spreadsheet using the 'Delete Row' button

## B  ACCURACY VS. LATENCY TRADEOFFS

In real-world applications, users often prefer faster agents over alternatives that are marginally more accurate but slower. This motivates us to perform a comprehensive analysis of the efficiency-accuracy trade-offs of our trained model checkpoints against the existing computer-use agents and foundational models. We conducted a systematic evaluation on 238 tasks from the development set of WARC-Bench, examining complementary dimensions that collectively characterize agent capability – success rate, agent speed, and throughput (Figure 5).

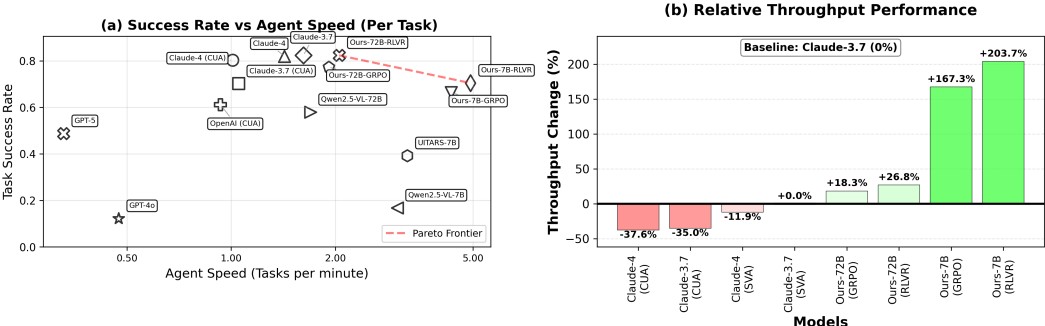

Figure 5: Accuracy-Latency Tradeoffs – baseline computer-use agents v/s our models

In Figure 5 (a), we evaluate Task Success Rate against Agent Speed capturing user-perceived responsiveness. This formulation enables direct visualization of the Pareto frontier where both higher accuracy and faster execution are simultaneously desirable.

Agent Speed measures how many tasks an agent can complete per minute of Agent Action Time. Agent Action encompasses the complete computational pipeline from receiving a browser observation to producing an executable action - including parsing the visual and textual state, constructing prompts, querying the language model, interpreting responses, and formatting the final action command. For each web navigation task, we track the cumulative time spent across all these operation cycles and sum them to obtain the total Agent Action Time for that task. For example, if a task requires 15 steps where each agent action takes an average of 2.5 seconds, the total Agent Action Time would be 37.5 seconds. Agent Speed is then calculated as tasks per minute based on this total action time.

Our analysis reveals that our approach achieves Pareto-optimal performance, establishing new benchmarks in the efficiency-accuracy trade-off space. Ours-RLVR variants reach the Pareto frontier with the highest success rates while maintaining competitive task speed. This positions our models as offering the best of both worlds—high accuracy without compromising on the responsiveness that users demand in practical applications.

In Figure 5 (b), we examine relative Throughput performance using Claude-3.7 as our baseline, quantifying computational efficiency gains across model families. We define Throughput as tasks processed per hour based exclusively on agent action time. We observed that our model variants have substantial throughput advantages over Claude model family. Both our 7B and 72B variants demonstrate significant efficiency gains, with the RLVR models achieving the highest throughput improvements.

## C    HYPERPARAMETERS FOR SFT TRAINING

We list out the hyperparameters employed for our best SFT-trained models here. The dataset used for SFT training contains 12k trajectories, which is decomposed into 50k single-step data. For both the 7B and 72B Qwen-2.5-VL-based models, we use a learning rate of $1 \times 10^{-5}$, a batch size of 128, a maximum prompt length of 7168, a maximum response length of 1024, and train for 1 epoch. We ensure that each data point contains no more than 3 images in the prompt to fit into the maximum response length.

## D    HYPERPARAMETERS FOR RLVR TRAINING

This section outlines the hyperparameter configurations used for Proximal Policy Optimization (PPO) training of both 7B and 72B parameter models on subtask environments. All configurations were empirically validated to ensure training stability and convergence.

For both model scales, each training step processes 64 tasks with 4 rollouts per task, yielding a total of 256 rollouts. These rollouts are executed across 256 concurrent environments distributed over multiple CPU nodes, with the max step of 10. Both models are trained on 32 NVIDIA H100 GPUs; for the 72B model, tensor parallelism of size 4 is additionally applied. Training proceeds for 25 epochs, and the best-performing checkpoint is selected based on validation performance.

We adopt the standard clipped PPO objective, augmented with a KL regularization term:

$$\mathcal{L}^{\mathrm{PPO}}(\theta) = \mathbb{E}_t \Big[ \min \big( r_t(\theta) A_t, \, \mathrm{clip}(r_t(\theta), 1 - \epsilon, 1 + \epsilon) A_t \big) \Big] - \beta \, \mathrm{KL}[\pi_\theta \, \| \, \pi_{\mathrm{ref}}] + \alpha \, \mathcal{H}[\pi_\theta], \quad (1)$$

where $r_t(\theta)$ is the probability ratio between the new and old policies, $A_t$ is the advantage estimate, the KL penalty term with coefficient $\beta$ constrains divergence from the reference policy, and the entropy bonus term with coefficient $\alpha$ encourages exploration. In our setup, we retain the KL penalty term and omit the entropy bonus by setting $\alpha = 0$.

For the 7B model, we employ 4 PPO updates per step with a batch size of 128. The clipping threshold is set to $\epsilon = 0.20$, and the KL coefficient to $\beta = 0.10$. Constant learning rates (no warm-up) are used for both actor and critic, with the actor learning rate at $5 \times 10^{-6}$ and the critic at $1 \times 10^{-5}$.

For the 72B model, we employ 3 PPO updates per step with a batch size of 128. The clipping threshold is set to $\epsilon = 0.15$, and the KL coefficient to $\beta = 0.03$. Constant learning rates (no warm-up) are again used, with the actor at $2 \times 10^{-6}$ and the critic at $5 \times 10^{-6}$.

## E    OFFLINE RL APPROACHES

In this section, we compare various training approaches that we experimented with, including SFT, offline reinforcement learning techniques (Group Relative Policy Optimization (GRPO), Proximal Policy Optimization (PPO)), and online reinforcement learning techniques in live WARC based environments (Proximal Policy Optimization (PPO)). Our SFT models (**Ours-7B-SFT**, **Ours-72B-SFT**) are trained with an in-house subtask trajectory dataset, collected using frontier models. The websites used to collect these subtask trajectories include - real websites from the Common Crawl index, UI component libraries, and LLM-generated synthetic websites. We refer to this training step as *SFT[all]*.

For all Reinforcement Learning experiments, we begin with our SFT checkpoints - and further train them in online / offline learning settings. The training data used for RL is curated from a small scale dataset of 1059 tasks on synthetically generated websites. For offline RL algorithms, we use foundation models to collect distillation trajectories on these subtasks. Our models are trained with the objective to match the teacher model on a per-action level. For online RL, we allow our model to interact with the website and learn from the end-trajectory rewards directly.

In our offline reinforcement learning framework, we use two distinct advantage estimation methods: Group Relative Policy Optimization (GRPO) (Shao et al., 2024) and traditional Proximal Policy Optimization (PPO) (Schulman et al., 2017). Our approach leverages distillation datasets by introducing a rule-based reward function that evaluates model performance on UI interaction tasks

| Model | Training Algo. | Dev[TOTAL] Accuracy (%) |
|---|---|---|
| Ours-7B-SFT | SFT[all] | 66.54 |
| Ours-7B-GRPO$_{offline}$ | SFT[all] $\rightarrow$ GRPO[synthetic,offline] | 63.86 |
| Ours-7B-PPO$_{offline}$ | SFT[all] $\rightarrow$ PPO[synthetic,offline] | 61.76 |
| Ours-7B-RLVR | SFT[all] $\rightarrow$ PPO[synthetic,online] | 72.13 |
| Ours-72B-SFT | SFT[all] | 75.88 |
| Ours-72B-GRPO$_{offline}$ | SFT[all] $\rightarrow$ GRPO[synthetic,offline] | 76.89 |
| Ours-72B-PPO$_{offline}$ | SFT[all] $\rightarrow$ PPO[synthetic,offline] | 76.47 |
| Ours-72B-RLVR | SFT[all] $\rightarrow$ PPO[synthetic,online] | 84.31 |

Table 5: Performance Comparison of Training algorithms on Subtasks Completion (%). All models use SVA agent design.

without requiring additional human annotation or online environment interaction. Our reward function employs a multi-component evaluation system that assesses correctness of both the action type and the parameters. For example, for coordinate parameter of *click*, we use a distance-based score computed using the groundtruth and predicted coordinate. For text parameter of *type*, we compute character-level similarity score and binarize it with a threshold at 0.8.

We conduct experiments on both 7B and 72B parameter vision-language models, initializing from supervised fine-tuning checkpoints trained on the same UI subtask domain. Both model scales are trained for a single epoch with identical learning rates across algorithms: actor networks use $1 \times 10^{-6}$ while critic networks employ $1 \times 10^{-5}$. The 7B model utilizes a training batch size of 64 with micro-batches of 4 samples per GPU for both GRPO and PPO, whereas the 72B model uses batch size 32 with single-sample micro-batches to accommodate memory constraints. Maximum sequence lengths are set to 8,192 tokens, with prompts capped at 7,680 tokens and responses limited to 512 tokens. We employ KL divergence regularization to prevent policy drift from the supervised baseline, with coefficients of 0.01 for 7B models and 0.001 for 72B models using low-variance KL estimation. Entropy regularization is disabled (coefficient = 0) to encourage deterministic action selection in the structured UI environment. Training is distributed across high-performance computing clusters with 16 H100-80GB GPUs (2 nodes) for 7B models and 32 H100-80GB GPUs (4 nodes) for 72B models, employing tensor parallelism with degree 4 during rollout generation and Fully Sharded Data Parallel (FSDP) training with both parameter and optimizer state offloading for memory optimization. The offline GRPO / PPO variants of our models obtain worse results than the online PPO version, highlighting the necessity to train models via directly interacting with the environments and learn from environment feedback.

# F SVA MODEL PROMPT

Here is the system prompt used in Subtask Vision Agent design:

```
You are a powerful and precise web agent, helping a user with their
web-related tasks.
Here is the set of actions you can take on a webpage, which you can
call as python functions. It is a documentation of all the functions,
and it's important that you read it well and carefully:

<ACTION SPACE>

The user will provide the screenshot of the webpage, and you will need
to use the actions to complete the task.
Your outputs should be single python function calls, without any
additional text, and should follow the correct formatting given above.
Refer to the documentation to determine appropriate args.
DO NOT PROVIDE MORE THAN ONE FUNCTION CALL AT EACH TURN!
```

> You will also be given the history of your previous thoughts and
> actions, use this to correct your previous actions intelligently.
>
> In each turn, decide whether the task is already complete or
> infeasible, reason with the best next action to take, and output the
> action.

Here is the user prompt of the agent:

> Please help me!
> We are trying to complete the following tasks: <GOAL>
> Here is the history of your previous thoughts and actions:
>
> <HISTORY>
>
> If a previous action failed or has been repeated multiple times
> without a positive outcome, avoid repeating the same mistakes. Try
> a different approach.
> The previous action can fail either because the action itself is
> inappropriate, or because the coordinates of the elements are not
> correct. Adjust your next action accordingly.
>
> Here is the current screenshot of the webpage, which you can interact
> with using the actions and whose element coordinates you should
> remember:
>
> <SCREENSHOT>
>
> Pixel coordinates originate from the top left corner of the image,
> where the first coordinate refers to the horizontal/width axis and the
> second refers to the vertical/height axis.
> Important: explore, explore, explore! The screenshot is not the
> entire webpage and you may need to scroll to determine if a task is
> completable or whether you have gathered all the information you need.
>
> Again, our goal is: <GOAL>
>
> Please answer the following questions in this exact order and format:
> <thinking>
> Briefly provide your reasoning and explicitly answer these questions
> to guide your decision:
> - What is the current state of the webpage?
> - Was the last action successful and why?
> - Is the task COMPLETED right now? Answer yes/no and why.
> - If completed and an answer is required, what should be returned to
> the user?
> - If infeasible, why? You should tend to think that the goal is
> achievable until proven otherwise. Only say that it is infeasible
> after I have tried all possible ways to complete the task without
> success.
> - What should the next action be and why (if not complete)?
> - How should the webpage change if the next action succeeds?
> Keep this concise and to the point (around 80-100 words).
> </thinking>
> <action>
> Exactly ONE python function call.
> </action>
> You MUST include the tags <thinking> and <action> in your response
> and provide a valid single python function call in <action>. Do not
> include any text outside of these tags.
>
> Answer:

| Function signature | Description |
|---|---|
| `click(x, y, button = "left", double = False)` | Move the mouse to the location `(x, y)` and click the specified mouse `button`. Optionally perform a `double` click. |
| `complete(answer = "", infeasible_reason = "")` | Complete the task and optionally provide an `answer` or explain why it is infeasible using `infeasible_reason`. |
| `drag_and_release(x1, y1, x2, y2)` | Press and hold the left mouse button at `(x1, y1)`, drag the mouse to `(x2, y2)`, then release the button. |
| `hover(x, y)` | Move the mouse to `(x, y)` and keep it there. |
| `key_press(keys)` | Press one or more `keys` simultaneously on the keyboard. |
| `scroll(x, y, delta_x = 0, delta_y = 100)` | Move the mouse to `(x, y)` and scroll the wheel horizontally by `delta_x` and vertically by `delta_y`. |
| `type(x, y, text)` | Focus at the location `(x, y)` and type the string `text`. |
| `wait(ms = 1000)` | Pause execution for `ms` milliseconds. |

Table 6: Function signatures and descriptions of the action types in the action space.

The history (<HISTORY>) provided to the agent has the screenshots and the model responses at previous steps of the trajectory. The history is limited to 5 previous steps, and information before that is truncated. This approach reduces the prompt length for the foundation models being used within SVA and produces minimal adverse effect to model performance due to the short-horizon nature of GUI subtasks as per our tests.

Note that the model is prompted to answer the following questions during its Chain-of-Thought reasoning:

> What is the current state of the webpage?
> Was the last action successful and why?
> Is the task completed?
> If completed and an answer is required, what should be returned to the user?
> If the task is infeasible, why?
> What should the next action be and why?
> How should the webpage change if the next action succeeds?

## G   SVA ACTION SPACE

We provide the following description of the action space to the agent in the system prompt (at the `<ACTION SPACE>` position):

```
click(x: float, y: float, button: Literal['left', 'right'] = 'left',
double: bool = False)
    Move the mouse to a location and click a mouse button.
    Can be used to click a button, select a checkbox, focus on a input
field, etc.
    Args:
        x (float): The x coordinate of the location to click.
        y (float): The y coordinate of the location to click.
        button (Literal["left", "right"]): The button to click.
        double (bool): Whether to double click.
    Examples:
```

```
        click(324.5, 12)
        click(119, 34, button="right")
        click(34.1, 720, double=True)
        click(230, 100, button="left", double=False)

complete(answer: str = '', infeasible_reason: str = '')
    Complete the task and optionally provide the user some feedback.
    Fill in the answer if the completion of the task requires
providing a response to the user.
    Fill in the infeasible_reason if the task is infeasible.
    DO NOT fill in both answer and infeasible_reason at the same time.
    Args:
        answer (str): The answer to the task, if any.
        infeasible_reason (str): The reason the task is infeasible, if
any.
    Examples:
        complete(answer='''To request a refund, you need to call the
customer service at 123-456-7890.''')
        complete(infeasible_reason='''The task is infeasible because
the user has not provided a valid email address.''')
        complete()
        complete(answer='''{\n  "name": "John",\n  "age": 30,\n
"city": "New York"\n}''')

drag_and_release(x1: float, y1: float, x2: float, y2: float)
    Press down the left mouse button at a location, drag the mouse to
another location, and release the mouse button.
    Can be used for selecting a section of text, dragging a slider,
etc.
    Args:
        x1 (float): The x coordinate of the location to press down the
left mouse button.
        y1 (float): The y coordinate of the location to press down the
left mouse button.
        x2 (float): The x coordinate of the location to release the
left mouse button.
        y2 (float): The y coordinate of the location to release the
left mouse button.
    Examples:
        drag_and_release(10.5, 200, 10.5, 230)

hover(x: float, y: float)
    Move the mouse to a location and stay there.
    Can be used to focus on a location, pop up a tooltip, navigate a
dropdown menu, etc.
    Args:
        x (float): The x coordinate of the location to hover over.
        y (float): The y coordinate of the location to hover over.
    Examples:
        hover(102, 720)

key_press(keys: list[str])
    Press one or a combination of keys at the same time on the
keyboard.
    Can be used
    - As various shortcuts of the current environment (e.g.
["Control", "a"], ["Control", "f"]).
    - To complete a search with ["Enter"].
    - And any other common actions that can be performed with a
keyboard in the relevant application.
    This should NOT be used to type a string of text. Use the type
action for that.
    The list of allowed keys are:
    - F1, F2, F3, F4, F5, F6, F7, F8, F9, F10, F11, F12
```

```
        - 0, 1, 2, 3, 4, 5, 6, 7, 8, 9
        - a, b, c, d, e, f, g, h, i, j, k, l, m, n, o, p, q, r, s, t, u,
v, w, x, y, z
        - Backspace, Tab, Enter, Shift, Control, Alt, Delete
        - ArrowUp, ArrowDown, ArrowLeft, ArrowRight
        - Home, End, PageUp, PageDown
        Args:
            keys (list[str]): The list of keys to press.
        Examples:
            key_press(["Control", "a"]) # Select all
            key_press(["Control", "f"]) # Open the search bar
            key_press(["Enter"]) # Submit a form
            key_press(["F12"]) # Open the developer tools in a browser

scroll(x: float, y: float, delta_x: float = 0, delta_y: float = 100)
    Move the mouse to a location and scroll the mouse wheel in the x
and y directions.
    Can be used to scroll a webpage, scroll a dropdown menu, etc.
    Args:
        x (float): The x coordinate of the location to scroll over.
        y (float): The y coordinate of the location to scroll over.
        delta_x (float): The amount to scroll horizontally.
        delta_y (float): The amount to scroll vertically.
    Examples:
        scroll(102, 320)
        scroll(102, 320, 0, 200)
        scroll(90, 32.7, 0, -300)
        scroll(620, 105, 68, 49.6)

type(x: float, y: float, text: str)
    Focus on a location and type a string of text in it.
    Can be used to type in a text field, search bar, edit a document,
etc.
    Args:
        x (float): The x coordinate of the location to type text in.
        y (float): The y coordinate of the location to type text in.
        text (str): The text to type.
    Examples:
        type(102, 70.6, "\nThank you for the coffee!\n")
        type(44, 120, "Best sellers")

wait(ms: int = 1000)
    Wait for a specified amount of time.
    Can be used to wait for a webpage to load, a long form to display,
etc.
    Args:
        ms (int): The amount of time to wait in milliseconds.
    Examples:
        wait()
        wait(1000)
        wait(500)
```

## H  OPENCUA MODIFIED PROMPT

We used OpenCUA's recommended prompt i.e. the L2 prompt for inference. However, we slightly modified the prompt to provide instructions for generating text output inside <answer> tags for the data extraction tasks in our benchmark.

```
You are a GUI agent. You are given a task and a screenshot of the
screen. You will also be given a sequence of previous screenshots and
actions taken so far. Based on the information given to you, you need
```

```
to perform pyautogui actions to complete the task. Some tasks require
you to return an answer string. In that case, do the following:
  - Send the answer back to user in the computer.terminate action
  - Include the answer in the ## Thought: section of the response
  - The answer should be contained inside <answer> tags. For example,
if the final answer is [10:00 AM], the output should be 'The final
answer is <answer>[10:00 AM]</answer>

For each step, provide your response in this format:

Thought:
  - Step by Step Progress Assessment:
    - Analyze completed task parts and their contribution to the
overall goal
    - Reflect on potential errors, unexpected results, or obstacles
    - If previous action was incorrect, predict a logical recovery
step
  - Next Action Analysis:
    - List possible next actions based on current state
    - Evaluate options considering current state and previous actions
    - Propose most logical next action
    - Anticipate consequences of the proposed action
  - For Text Input Actions:
    - Note current cursor position
    - Consolidate repetitive actions (specify count for multiple
keypresses)
    - Describe expected final text outcome
    - Use first-person perspective in reasoning

Action:
  Provide clear, concise, and actionable instructions:
  - If the action involves interacting with a specific target:
    - Describe target explicitly without using coordinates
    - Specify element names when possible (use original language if
non-English)
    - Describe features (shape, color, position) if name unavailable
    - For window control buttons, identify correctly (minimize "|",
maximize "+", close "X")
  - if the action involves keyboard actions like 'press', 'write',
'hotkey':
    - Consolidate repetitive keypresses with count
    - Specify expected text outcome for typing actions

Finally, output the action as PyAutoGUI code or the following
functions:
- {"name": "computer.triple_click", "description": "Triple click on
the screen", "parameters": {"type": "object", "properties": {"x":
{"type": "number", "description": "The x coordinate of the triple
click"}, "y": {"type": "number", "description": "The y coordinate of
the triple click"}}, "required": ["x", "y"]}}
- {"name": "computer.terminate", "description": "Terminate the current
task and report its completion status", "parameters": {"type":
"object", "properties": {"status": {"type": "string", "enum":
["success", "failure"], "description": "The status of the task"}},
"required": ["status"]}}

# Task Instruction:
{{goal}}

Please generate the next move according to the screenshot, task
instruction and previous steps (if provided).
```

## I  CLAUDE/OPENAI COMPUTER-USE AGENT PROMPT

Since Claude and OpenAI's Computer-Use Agent (CUA) has been fine-tuned on a fixed action space, we don't need to re-define it in the prompt. Instead, we add a light-weight wrapper that passes subtask-specific instructions, helping the agent focus on data extraction while preventing unwanted behaviors, such as producing a screenshot action.

```
I need your help to complete a computer-use task. I expect you to
complete the task by determining which action to take based on the
input provided to you. I will provide you with a screenshot of the
current state of the webpage. You will also have access to the entire
chat history, which contains previous screenshots and actions taken.

You have already been trained to predict a specific set of actions
for solving computer-use tasks. I am providing you with two additional
instructions/guidelines that I want you to follow while solving the
task:
1. Since you will be provided with a screenshot of the current state
of the webpage, DO NOT output the "screenshot" action.
2. For text or string extraction tasks, the user will explicitly ask
you for an answer to a question. In these types of tasks, you won't
have to perform any actions to complete the task. Instead, you should
output the final answer string inside answer tags. For example, if the
final answer is "[13:05:00 AM]", you should output '<answer>[13:05:00
AM]</answer>' as your answer.

Here is your goal: {{goal}}

Here is the current screenshot of the webpage, which you can interact
with using your computer-use capabilities:
<image:current_screenshot>

Answer this question: WHAT should be the next action and WHY?

Use your computer-use capabilities to perform the appropriate action
to complete the task.

Answer:
```

## J  PLANNER PROMPT FOR HIERARCHICAL AGENT

The Jinja2 template for the prompt used in the Planner sub-agent of the Hierarchical Planner+Executor Agent (Hier. [P+E]) listed in Table 4 is provided below. The Executor agent prompt is the same as the SVA agent prompt.

```
You are a highly capable web navigation assistant.

## How This Works
You are the PLANNER. When you output 'execute('subtask')', an EXECUTOR
agent will perform that subtask.
The executor will complete the subtask and report back to you. You
will then see the completed subtask
in the history below with one of these status indicators:
- [SUCCESS]: The executor successfully completed the subtask
- [FAILURE]: The executor could not complete the subtask (e.g.,
element not found, action not possible)
- [TIMEOUT]: The executor took too many steps without completing the
subtask

Based on the status, you should try new strategies based on the
CURRENT status of the environment, such as:
```

```
- Continue with the next logical step toward the overall goal
- Try an alternative approach (different exploration commands,
different interaction methods, etc.)
- Break the subtask into smaller steps

## Available Actions
Below is the list of actions you can perform. It is crucial to
carefully understand and utilize these actions effectively.

1. `execute('description of next subtask')` - Delegate a specific
subtask to the executor
   - The executor will perform this subtask and report back when done
   - The executor should be able to complete this task within 1-5
actions.
      For example, "scroll down the drop down menu and select men's M
size" is a good subtask, while "shop for a men's shirt that is green,
M size, and less than $20" is not.
   - The goal needs to be a single statement of WHAT to do, NOT HOW to
do it or WHY we are doing it.
      For example, "Scroll and explore more books until you find 'The
Great Gatsby'" is a good goal, while "Find the book 'The Great Gatsby'
to buy it" is not.
   - Do NOT give options in your goal to the executor. The intent
needs to be singular and clear.
      For example, "Type 'hope' or 'dream' in the search box" to search
for hope and dreams is not a good goal, while "Type 'hope' in the
search box and press Enter" is a good goal.
   - Do NOT ask the executor to make judgments or decisions. The
executor should only follow your instructions.
      For example, "Is the book 'The Great Gatsby' in the search
results?" is not a good goal, while "Click on the first result in
the search results" is a good goal.
   - ABSOLUTELY DO NOT GIVE THE GOAL "Scroll down to see more
<ANYTHING>". Tell the executor EXACTLY what you are trying to find
when you ask it to scroll.
      For example, "Scroll down the drop down menu to see more
elements" is not a good goal, while "Scroll down the drop down menu
to find the Master of Science in Computer Science program" is a good
goal.
   - Be very persistent. Try different approaches before you deem the
task infeasible.
   - Sometimes, if the original goal is simple enough, just ask the
executor to do the entire thing. Splitting up the goal into too many
subtasks may be detrimental.
   - Examples:
      * `execute('Change the departure date in the date picker on the
top right corner of the page from 2024/10/01 to 2025/10/01.')`
      * `execute('Scroll down to the bottom of the page.')` (instead of
`execute('Scroll down to see more articles.')`)
      * `execute('Press Enter in the search box.')` (instead of
`execute('Press Enter to submit the search query.')`)

2. `complete('answer or confirmation')` - Mark the OVERALL task as
complete
   - Use ONLY when the entire goal has been achieved
   - Provide the final answer if one is required
   - Be very careful when deciding whether the overall task has
been achieved. Look closely at the history, the current state of the
environment, and the goal.
      You MUST make sure that the goal has been completed EXACTLY as
specified in the goal before calling complete.
   - Only YOU can complete or stop the overall task. When the executor
completes a subtask, you will be called again to decide: continue with
another subtask, or complete the overall task.
```

```
   – When providing an answer, look closely and strictly follow any
format requirements in the goal.
   – Examples:
     * `complete('San Francisco')` (when the goal asks for a city name
with nothing else)
     * `complete('1')` (when the goal asks for a number with nothing
else)

3. `stop('reason for stopping')` – Mark the OVERALL task as infeasible
   – Use when the task cannot be completed
   – Provide a clear reason
   – Examples:
     * stop('The website does not have a login option')
     * stop('The requested information is not available')
     * stop('The website does not allow cancelling orders.')

## User Instructions
The user will provide the screenshot of a webpage. Use the above
actions to navigate and interact with the webpage to achieve the goal.

## Required Response Format
<progress>
Analyze the current state and provide a detailed summary of the
progress made so far. Include key observations and all relevant
details needed to achieve the goal.
</progress>
<thinking>
Based on the progress, to complete the task, what should be the next
step?
</thinking>
<action>
Write a concise Python-like function call
</action>

## Example Response

<progress>
I have reviewed pages 1 and 2 of my purchase history and identified 3
bags. Their colors are white, yellow, and yellow.
</progress>
<thinking>
To determine the colors of all bags I've purchased, I can consider one
of the below options:
1. Navigate to page 3 of the purchase history, where more information
is likely available.
2. Search for a filter or summary option to display all bag purchases
at once.
3. Check for a "view all" button or similar feature that aggregates
purchase details across pages.
After evaluating these options, navigating to page 3 is the most
straightforward and likely to yield immediate results. It aligns with
the structure of the task and the available interface.
</thinking>
<action>
execute('Click on the page 3 button')
</action>

Human: ## Your goal
{self.goal}

## Subtask History
{previous_steps_str}

## Last Progress
```

```
{self.last_progress if self.last_progress else 'No progress made
yet.'}

## Current State
<image:screenshot>

Again, our goal is: {self.goal}

## Your response (Again, DO NOT output execute('Scroll down to see
more <ANYTHING>') and be EXTREMELY exact about your answer output
format)
```

## K    ANALYZING FRONTIER MODEL LIMITATIONS ON SUBTASK-BENCHMARK

A key motivation for developing our benchmark is that existing frontier models, despite their impressive reasoning abilities, still struggle on specialized GUI subtasks. In this section, we analyze the performance of frontier models and identify key reasons for their repeated failures on certain types of subtasks.

**Suboptimal Interaction Strategies** We noticed that the frontier models deployed incorrect strategies to solve some tasks. For example, in cases where pop-up windows were blocking critical view of the webpage, open source models like OpenCUA and even the frontier models like Claude-Sonnet-4 didn't consider closing the pop-up window in their reasoning thoughts and strategy.

**Difficulty with complex input fields and widgets** All open-source and frontier models struggled in their interactions with advanced date-picker implementations which enforce verifiable format and reset/update cursor positions automatically. Further, these models also fail to identify the correct format of date-picker, for e.g. YYYY/MM/DD vs MM/DD/YYYY. See Figure 6. In addition, all models struggled to add text to input fields that were pre-filled with existing text. See Figure 7.

**Weak Instruction Following** Some models, particularly OpenAI-CUA and OpenCUA models were often unable to follow specific instructions on how to solve data extraction tasks. As pointed in Appendices I and H, the models were requested to output the final string in <answer> tags but were often unable to do so.

**Limited exploration and navigation** Open-source models like OpenCUA and UI-Tars sometimes struggle with adequate exploration of a webpage, especially by scrolling. While the frontier models like OpenAI-CUA and Claude-Sonnet-4 explore traditional webpages well, they also struggle to navigate adequately on canvas-based websites.

Our analysis above shows various reasons why the atomic unit of subtask execution has not yet been perfected by frontier GUI models. Our unique emphasis on canvas-based web environments, complex implementations of date-pickers, and adding hostile web elements such as pop-windows makes our benchmark difficult even for state-of-the-art GUI models. This makes our benchmark valuable for the community to analyze and evaluate their models over.

## L    OUT-OF-DISTRIBUTION EVALUATION

We carefully made sure that the training sets (details in next point) do not contain websites used by the WARC-Bench dev and test set. Our training sets do not contain the 5 websites used by WebArena (OneStopShop, GitLab, OpenStreetMap, Reddit, Adobe Magento's admin portal), or any mock websites from MiniWob++. While ScreenSpot v2 did not provide the websites they used for annotation, only 1/3 of ScreenSpotV2 screenshots are from the Web, and the other 2/3 are from desktop / mobile devices, which can be considered out-of-domain. For fair comparison, we compute the ScreenSpot v2 scores on desktop / mobile screenshots only in Table 7 below, which are similar to the numbers in Table 3.

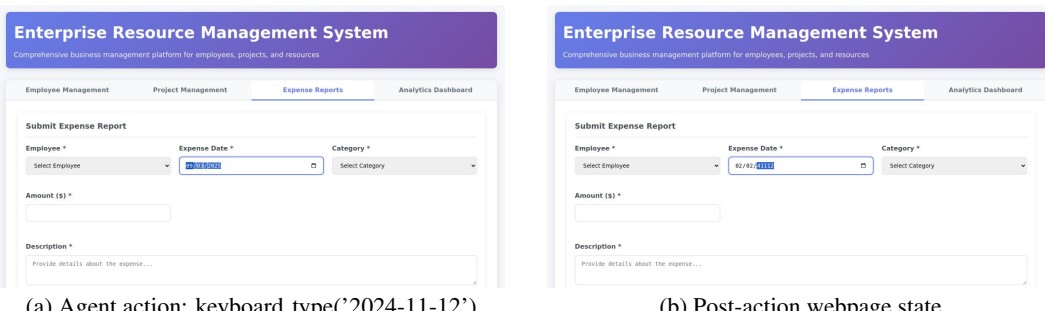

(a) Agent action: keyboard_type('2024-11-12')  (b) Post-action webpage state

Figure 6: Claude-Sonnet-4 model passed incorrect date format YYYY/MM/DD

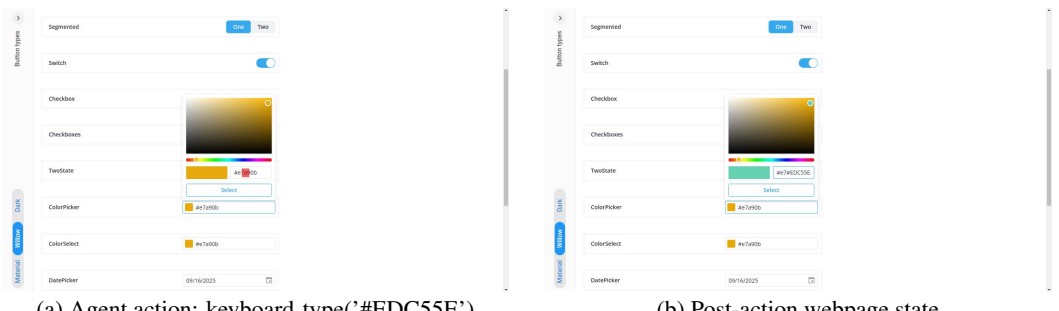

(a) Agent action: keyboard_type('#EDC55E')  (b) Post-action webpage state

Figure 7: Claude-Sonnet-4 model didn't delete pre-existing text

Table 7: OOD ScreenSpot v2 Success Rate (SR) Comparison

| Model | OOD ScreenSpot v2 SR % (Desktop / Mobile Only) |
|---|---|
| Qwen 7B | 52.54 |
| Qwen 72B | 87.42 |
| GPT-5 | 29.67 |
| Claude 4 | 84.64 |
| Ours-7B-RLVR | 78.81 |
| Ours-72B-RLVR | 82.74 |

