# OpenReview forum: "WARC-Bench: Web Archive based Benchmark for GUI Subtask Executions"
_ICLR.cc/2026/Conference — ICLR 2026 Poster_

### Official Review · Reviewer_pxfP · 2025-10-20

**Soundness:** 3
**Presentation:** 4
**Contribution:** 2
**Rating:** 4
**Confidence:** 4

**Summary:**

This paper introduces WARC-Bench, a new benchmark on short-horizon web browsing tasks. It is somewhere between single-step visual grounding tasks and sophisticated, long-horizon web interaction tasks. They also developed a way to synthesize more of such tasks. They evaluated multiple computer-use agents and their own Subtask Vision Agent (SVA), using frontier models, open-weights models, and custom models post-trained with SFT and RLVR.

**Strengths:**

1. Innovative way to scale the size of the dataset, including the use of WARC files to quickly reproduce exact replication of complex real-world websites.

2. A good methodology to synthesize data and demonstration to show the effectiveness of training on such datasets.

3. Strong results of the authors' own trained model. Discussion on behavioral analysis of SFT v.s. RLVR is insightful.

**Weaknesses:**

1. Baseline only used computer-use agents but not web browser agents (e.g. browser-use) or general agents with web browsing capability (e.g. OpenHands).

2. Lack of comparison to other short-horizon browsing benchmarks, e.g. WebGames, WebVoyager, WebShop.

3. Experiments are run 3 times but there's no error bar or other variation info.

4. No human baseline. Since these are short-horizon subtasks, it seems reasonable for human baseline to reach 100%. Lack of human baseline makes it hard to evaluate the soundness of the benchmark (e.g. whether there are ambiguous instructions, wrong answers, cheating possibilities, etc.).

5. Dev accuracy on synthetic dataset is universally higher than dev accuracy on real dataset, indicating the synthetic dataset might have design flaws (e.g. data contamination, overly simple designs, etc.).

**Questions:**

The definition of a "subtask" seems somewhat arbitrary. Authors argue this is a "novel" benchmark designed to evaluate AI agents in executing "subtasks" on web browsers. How are these "subtasks" different from the tasks in similar benchmarks like WebGames, WebVoyager, WebShop?

---

> ### Author Response · Authors · 2025-11-25
>
> We thank the reviewer for their insightful comments.
>
> ## W1: Baselines with web browser agents or general agents
>
> We were unable to find an easy way to connect OpenHands with WARC-Bench environments, which require connection over Chrome DevTools Protocol, hence we only provide the results for Browser-Use on WARC-Bench datasets in the table below.
>
> | Model | Dev-Real | Dev-Synthetic | Test |
> |---|:---:|:---:|:---:|
> | CUA (w/ claude-sonnet-4-20250514), Table 2 | 76.11% | 79.92% | 47.17% |
> | SVA (w/ claude-sonnet-4-20250514), Table 2 | 81.67% | 84.27% | 64.83% |
> | Browser-Use (w/ claude-sonnet-4-20250514), _New_ | 85% | 88.27% | 57.5% |
> Please note,
> 1. The Browser-use agent design uses both axtree and screenshot-based grounding to execute actions while the agents discussed in the paper are all vision-only agents.
> 2. We were only able to get 1 evaluation run per split for Browser-use due to time and resource constraints.
> 3. We conclude that complex and iterative agent designs like Browser-use (adaptive observation space, prompting optimizations etc.) can further increase performance for these models, but we have not focused on that in our paper and stick to a simple SVA design.
> 4. Dev-Synthetic split contains LLM generated axtrees which are a lot shorter and cleaner than real life website HTMLs. Thus, a high performance on this split by a text based agent is expected.
>
>
> ## W2: Comparison to WebGames, WebVoyager, WebShop
>
> WebGames (https://webgames.convergence.ai/) contains 50 interactive UI challenges which are synthesized by the authors, but are not realistic websites. WebVoyager collects 643 tasks based on 15 live common websites, and only 22.3% of the tasks have stable golden answers, while others might have open-ended answers or depend on the live state of the websites. On the contrary, WARC-Bench contains precise evaluation rules for each task which can guarantee stable evaluation. WebShop focuses on shopping tasks on a simulated e-commerce website only. In comparison, WARC-Bench consists of 29 diverse web environments spanning domains like healthcare, CRM, enterprise, and developer websites (Github, Kaiser, Zendesk, Scrimba etc). WARC-Bench is scalable to even more by domains by design. We believe WARC-Bench makes new contributions by providing a realistic and stable subtask-focused evaluation set, as well as a training set which supports online RLVR and is usually absent in previous benchmarks.
>
> ## W3: Variation info of 3 eval runs
>
> Thank you for pointing this out. We have added the standard deviations to Table 2 for the scores on the development set and the test set, except for OpenCUA 7B/32B which was run only 1 time (originally indicated in the table caption), as they were expensive and slow to run via Huggingface Inference APIs (the only option to host the OpenCUA model at the time due to missing VLLM support).
>
> https://huggingface.co/xlangai/OpenCUA-7B#vllm-support
>
> ## W4: Human baseline
> We did human evaluation on the 60 manual tasks in the development set (Dev-Real) using 3 humans, with 1 minute maximum per task. The persons who did the evaluation did not participate in the dataset creation. The human accuracy we got is **87.11% +- 1.84% (52.3/60 tasks)**.
>
> Among the failed tasks, tasks requiring reading long text or formatted outputs are most challenging for humans. For example, they failed the towing fee extraction task where one needs to scan through rules / tables of fees to come up with the final number. For tasks that require formatted outputs, sometimes the answers were written in slightly wrong format. We will conduct more human evaluations and include the results in the paper.
>
> ## W5: Accuracy on synthetic data is higher than realistic data
>
> We acknowledge that the synthetic data split is simpler than the manually collected data split, as they are auto generated by frontier models. This is why we did not include any synthetic data in the test set. The reason manual tasks are more complex is due to the visual complexity (pages have densely packed HTML elements) and dynamism (they have pop-ups, tooltips, error messages etc.) of real websites compared to synthetic ones. The more complex/dynamic real website could indeed lead to humans creating more complex subtask goals too – but that is not by design. We believe synthetic data quality can be further improved as the coding quality of frontier models keeps becoming better.

---

> > ### Comment · Reviewer_pxfP · 2025-11-25
> >
> > I appreciate the added experiments and human baseline. Although I am still not fully convinced of the significance of the contribution due to the narrow definition of "subtasks," I like the presentation and the methodology. I have raised the score accordingly.

---

### Official Review · Reviewer_f6t5 · 2025-10-31

**Soundness:** 2
**Presentation:** 3
**Contribution:** 3
**Rating:** 4
**Confidence:** 3

**Summary:**

- This paper introduces WARC-Bench, a comprehensive training and evaluation dataset to train and evaluate models’ and agents’ ability to complete GUI sub-tasks in web environments using computer-use agents.
- The authors implement an agentic scaffold (subtask vision agent: SVA) for inference and training and train 7B and 72B LLMs using supervised fine-tuning on trajectories sampled from stronger LLMs, followed by online RL (using PPO) on the synthetic training data.
- Experimental results demonstrate the effectiveness of their approach on the test split of WARC bench while also generalizing to other web navigation tasks/benchmarks.

**Strengths:**

- The paper is well-written and easy to understand.
- The task of completing specific sub-tasks on webpages is an interesting method of training models for web navigation tasks.
- The authors provide extensive comparisons across open- and closed-source models, showing improvements through supervised fine-tuning and RLVR.

**Weaknesses:**

- Missing baselines for other web navigation datasets: For WebArena, can you please include a comparison of your work with AutoGLM (https://dl.acm.org/doi/abs/10.1145/3637528.3671620), NNet-Nav (https://openreview.net/forum?id=L5BYl0GE2F) and Go-Browse (https://arxiv.org/abs/2506.03533) that also develop training data and train open-source LLMs as web agents. In particular, I am not convinced about the claim of your 7B model generalizing to WebArena benchmark because Go-Browse reports a resolve rate of 8.3% on WebArena with Qwen-2.5-7B-Instruct, which is slightly better/comparable to the performance of your fine-tuned 7B model.
- Generalizability of results: What are the specific websites used during training and are they semantically similar to the websites used in the evaluation benchmarks, particularly WARC-bench test set, WebArena, MiniWob++, and ScreenSpotV2. It would be helpful if authors could include additional discussion on out-of-domain generalization. For example, if the training set does not include any task dealing with shopping websites, can the trained models solve such tasks in the evaluation split? Relatedly, is there any analysis on how the trained models generalize to subtasks not seen during training? For example, if the training dataset does not include subtasks required to fill forms and manipulate tables, can it solve these tasks during inference?
- Exact training dataset used for SFT: What are the specific tasks and websites used during the supervised fine-tuning stage? The paper initially mentions the training split of 1059 instances in the WARC bench, but then describes using 12k unique trajectories using teacher models from Common Crawl, and synthetic data? How are the natural language descriptions of tasks designed for these 12K trajectories?

**Questions:**

Please address each of the weaknesses.

---

> ### Author Response · Authors · 2025-11-25
>
> Thank you for your thoughtful review. We address the specific queries below:
>
> ## W1: Missing baselines for other web navigation datasets
>
> Our aim is to not evaluate models for long horizon planning and web navigation tasks like WebArena (WA). Our model was trained to specifically be good at subtasks and not long-horizon tasks. We were unable to find a model checkpoint for AutoGLM and hence, were unable to benchmark it.
>
> 1. Both NNetNav (16.3% on WA) and Go-Browser (21.7% on WA) were trained on trajectories collected from WA environments, thus making the comparison with our 7B model (7.3% on WA) unfair, since both WA environments as well as their task complexity is out of distribution for our model.
> 2. Both NNetNav and Go-Browse use text based observation spaces (HTML/axtrees) and our paper limits its focus to vision-only agents due to space and resource constraints.
> 3. Table 3 shows that learning subtask execution is useful for improving long horizon task solving in WebArena, but we do not expect our agents/models to beat other models/agents on long-horizon web navigation benchmarks.
> 4. We run the Go-Browse and NNetNav models on our benchmark and provide additional results below. Results are reported over 3 runs. The table shows that good performance on WA doesn't necessarily translate to WARC-Bench, proving that there remains an unaddressed gap in existing benchmarks, that WARC-bench fills.
> | Model | WebArena SR % | WARC-Bench SR % [Dev-Real] | WARC-Bench SR % [Dev-Synthetic] |
> |---|:---:|:---:|:---:|
> | NNetNav (8B) | 16.3 | 2.2 ± 0.96 | 6.74 ± 1.12 |
> | Go-Browse (7B) | 21.7 | 15.53 ± 0.96 | 21.53 ± 0.32 |
> | Ours-7B-RLVR | 7.31 | 54.44 | 78.09 |
>
> ## W2: Additional discussion on out-of-domain generalization.
>
> **Concern: “What are the specific websites used during training and are they semantically similar to the websites used in the evaluation benchmarks”**
>
> We carefully made sure that the training sets (details in next point) do not contain websites used by the WARC-Bench dev and test set. Our training sets do not contain the 5 websites used by WebArena (OneStopShop, GitLab, OpenStreetMap, Reddit, Adobe Magento’s admin portal), or any mock websites from MiniWob++. While ScreenSpot v2 did not provide the websites they used for annotation, only 1/3 of ScreenSpotV2 screenshots are from the Web, and the other 2/3 are from desktop / mobile devices, which can be considered out-of-domain. For fair comparison, we compute the ScreenSpot v2 scores on desktop / mobile screenshots only in the table below, which are similar to the numbers in Table 3.
>
> | Model | OOD ScreenSpot v2 SR % (Desktop / Mobile Only) |
> |---|:---:|
> | Qwen 7B | 52.54 |
> | Qwen 72B | 87.42 |
> | GPT-5 | 29.67 |
> | Claude 4 | 84.64 |
> | Ours-7B-RLVR | 78.81 |
> | Ours-72B-RLVR | 82.74 |
>
> **Concern: “How do the trained models generalize to subtasks not seen during training?”**
>
> For generalization of unseen subtasks, we did an initial analysis and found that the model would need to see relevant examples to be able to perform tasks unseen or uncommon in the offline training set. For example, the “sorting” tasks require the model to click on the column header to sort the column in ascending or descending order, and the offline training set contains little such examples. The SFT’ed model failed this task and often assumed the sorting order from the arrow direction but not the actual sorted results. On the other hand, the RLVR model was able to learn the correct sorting behavior based on sorted results via interacting with the environment. We will conduct more analysis and include it in the paper.
>
> ## W3:  Exact training dataset used for SFT
>
> **Concern: “What are the specific tasks and websites used during the supervised fine-tuning stage?”**
>
> We used two training sets: an offline training set with subtask trajectories on randomly sampled websites from CommonCrawl (used for SFT training) and the WARC-Bench training split contains only online environments (used for RLVR training after SFT). Here are all the steps we did to curate the CommonCrawl SFT trajectories:
>
> 1. LLM based goal generation: The input to the pipeline is a list of URLs we sample from Common Crawl. During this process, a simple LLM agent navigates to each website, collects basic information such as HTML and screenshots, decides whether the website should be included in our dataset based on several factors, such as safety and complexity. Based on the HTML and screenshots, LLM is prompted to generate a set of completable subtasks over the website. We will release the exact goal-generation prompt with the code release since it is fairly long.
> 2. Trajectory Collection: We use a teacher model to solve these tasks on websites using the SVA agent design we outlined in the paper. During this process, we gather the agent trajectories of the teacher model. These Common Crawl trajectories are then filtered and processed to create our SFT dataset and used to train the SFT models.

---

### Official Review · Reviewer_iqHj · 2025-11-01

**Soundness:** 1
**Presentation:** 3
**Contribution:** 2
**Rating:** 2
**Confidence:** 4

**Summary:**

This paper introduces WARC-Bench, a new benchmark for evaluating GUI agents on short-horizon subtasks. It includes 438 tasks across 29 real and 62 synthetic websites, with deterministic programmatic evaluators (e.g., JS/URL/string matchers) and sandboxed replay environments. The paper also introduces Subtask Vision Agent and reports results across models. The authors further train Qwen-based models via  SFT and RLVR.

**Strengths:**

- The paper identifies an underexplored middle ground between visual grounding (single-step) and long-horizon navigation (multi-page workflows), positioning subtasks as a meaningful intermediate capability.

**Weaknesses:**

- Over 70% of training and development data is synthetic. While the paper acknowledges this, it remains unclear how well these synthetic subtasks capture the diversity and ambiguity of real-world web UIs. The manual verification process is under-detailed.


- The paper champions the high fidelity of WARC files but fails to adequately address their most significant drawback: they are static snapshots. A vast category of real-world web tasks involves interacting with dynamic, real-time, or session-dependent content (e.g., checking live stock prices, booking the last available seat on a flight). By design, WARC-Bench cannot evaluate an agent's ability to handle this dynamism. While the authors briefly mention limitations, they understate how this core architectural choice constrains the benchmark to a specific, offline subset of web interaction, making its claim of testing real-world capabilities only partially true.

- The paper frames "GUI subtasks" as a key conceptual contribution. However, this appears to be more of an incremental refinement than a novel problem class. Benchmarks like MiniWoB++ have long focused on widget-based interactions in simplified environments. WARC-Bench can be seen as a high-fidelity, WARC-based instantiation of this existing idea rather than a new paradigm. The contribution is therefore more on the engineering side (the use of WARC).


- The paper suggests that adding new environments is as simple as "adding a new web recording." This overlooks the fact that the primary bottleneck in creating meaningful benchmark tasks is the manual and intellectually demanding process of defining a clear, unambiguous goal and, crucially, authoring the programmatic reward function (the "evaluator"). This manual effort is not eliminated by the WARC approach. Therefore, the claim that the benchmark design is uniquely scalable compared to others that also require manual task specification is overstated.

**Questions:**

see above

---

> ### Author Response · Authors · 2025-11-25
> **W1: Data Concerns, W2: WARC containing static snapshots**
>
> **Concern: Over 70% of training and development data is synthetic, unclear how well these synthetic subtasks capture the diversity and ambiguity of real-world web UIs.**
>
> We appreciate this concern and would like to clarify our three-tier data strategy. Our benchmark deliberately involves 3 splits to balance scalability with experimental rigor:
> 1. Test set (200 tasks): 100% real world websites
> 2. Development set (238 tasks): 25% real and 75% synthetic websites
> 3. Training set (1059 tasks): 100% synthetic websites for scalable RL with verifiable rewards
>
> We provide visual evidence of the data quality [HERE](https://amazing-baklava-78fa98.netlify.app/) showing that our synthetic environments contain realistic and diverse UI widgets. The quality of the synthetic datasets is also validated through multiple empirical findings:
>
> 1. **Transfer validation:** Table 2 shows that RLVR training on synthetic data improves performance on both synthetic (+9.41%) and real (+7.78%) dev tasks. This result demonstrates the generalization from synthetic tasks to real websites.
> 2. **Score Correlation:** Agent performances in Table 2 show that the development set scores correlate well with the test set scores. The test set is 100% manually created.
> 3. **Diversity by design:** Figure 2 shows the subtask type distribution between test (100% real) and dev sets (mostly synthetic). Our synthetic task generation pipeline takes in the desired subtask type (form-filling, table manipulation, date picking etc.) to create in the generated websites. Thus, the synthetic dataset has good coverage of the various subtask types by design.
>
> **Concern: Manual verification process is under-detailed**
>
> We apologize for the lack of clarity. Our subtask collection involves 2 distinct processes:
> For synthetic tasks (dev-synthetic / train):
> - An LLM is prompted to generate HTML pages with desired widgets, task goals, and evaluator functions.
> - Then a human verifier opens the website, performs the goal on it, and checks whether the LLM-generated evaluator is correct or not.
> - In case of errors, the evaluator is either fixed, or the datapoint is discarded in case of an infeasible goal.
> - Time cost: **<1 min per task**
>
> For real tasks (dev-real / test):
>
> - Production websites are manually selected and recorded using WebRecorder based tool.
> - A human hand-writes the goals based on feasible and interesting subtasks on the website,
> - A human also finds the evaluator function. Coding a JS evaluator function is the most time consuming amongst the various evaluator function types.
> - Time cost: **~5-10 mins for recording, 5-10 minutes for task creation**
>
> Since synthetic data generation circumvents all the manual steps except for a simple verification, they are a lot more scalable to create.
>
> **Concern: WARC files are static snapshots... cannot evaluate an agent's ability to handle dynamism... making its claim of testing real-world capabilities only partially true.**
>
> We respectfully disagree with this concern. WARC files record real interaction sessions and capture all the information from the websites, including HTML, CSS and JavaScript codes and images. This information can be replayed in WARC-Bench to simulate the website page completely in a browser with mocked API return values, while preserving complex and dynamic JS features like wigdet functionality (interactive calendars, dropdowns, sliders), JS executions (form validation, animations), dynamic content rendering (popups, autocomplete). This provides stable environments without having to worry about changes on live websites which might invalidate the evaluation rules.
> While the WARC does not provide “live” server state (for checking real-time stock prices, real ticket prices), this is not required for testing widget interaction and subtask execution capability of web agents; since subtasks only requires client-side UI manipulation skills and not real-time data retrieval. Furthermore, the archived web states and simulated API calls can faithfully recover the level of influence real-time data has on the subtasks during recording.
> We want to explicitly clarify that we do not aim to build a general web agent benchmark (where live integration would be necessary).
>
> We are isolating and measuring widget interaction capabilities—a client-side skill that requires interactivity but not server state. Some WARC based websites are shown here that demonstrate how this can be achieved in WARC based websites [on this link](https://amazing-baklava-78fa98.netlify.app/).
>
> We have the following evidence that WARC based environments still pose genuine challenges to GUI agents:
> 1. Success rates vary by 27 percentage points across models (Table 2)
> 2. Frontier models still struggle to complete the tasks (best performance of 64.8% for Claude-4.0-Sonnet)
> 3. Error analysis (Appendix J) reveals systematic failures in complex interactions (date-picker formats, pre-filled fields, canvas-based UIs)

---

> ### Author Response · Authors · 2025-11-25
> **W3: Research Contribution / Novelty Concerns, W4: Scalability of Manual Data Collection**
>
> **Concern: "The paper frames "GUI subtasks" as a key conceptual contribution.... can be seen as a high-fidelity, WARC-based instantiation of this existing idea rather than a new paradigm."**
>
> Thanks for the callout! We agree that we are not inventing the concept of subtasks and that Miniwob++ is another benchmark with tasks of subtask-level complexity. We have edited L092 in the paper to reflect this better.
> However, through the establishment of much more high resolution and realistic website environments than those of Miniwob++, we have focused our dataset to both test and train model capabilities for agents that decompose real web tasks to meaningful sub-components. Miniwob++ tasks are geared more towards testing a model’s general reasoning capability or rudimentary understanding of UI.
>
> 1. __Realistic tasks:__ Consider, for example, the Miniwob++ tasks of drawing a line to bisect an angle, guessing a random number through trial and error, and clicking an item on a simplified menu. In comparison, our aim is to collect subtasks that realistically appear in web agent workflows (please find samples in Appendix A).
> 2. __Usefulness in real web workflows:__ Table 3 shows that the performance of GPT-5 and Qwen 72B on MiniWob++ and ScreenspotV2 do not represent the two models’ real web task solving capability as tested by WebArena. Meanwhile, our benchmark is better correlated with WebArena. This shows that high-fidelity subtasks can better predict long-horizon performance, and this is a unique contribution that WARC-Bench makes.
> 3. __Predictive Power:__ Table 3 shows that WARC-Bench provides more granular diagnosis of capability differences compared to WebArena. Delta between Qwen2.5VL 7B and Claude-4-0 Sonnet is 60% on WARC-Bench and only 34% on WebArena. This gap enables targeted model improvement - Appendix J shows how we can find ways to improve model capabilities on specific subtasks like date-pickers, input fields when using WARC-Bench.
>
> While we agree that we see the recording and replay of WARC as a key engineering contribution, we claim that our contribution goes above an engineering one. Besides the dataset contribution related to GUI subtasks, there are also several analytical studies we have done that were included in this paper.
>
> 1. Effect of RLVR training (Section 5): Models improve via enhanced visual grounding (-0.94 steps/task), better exploration (+23% scrolling), and reduced redundancy.
> 2. Validating transfer learning: RLVR on synthetic web environments can increase model performance by 7.78% on production grade websites.
> 3. Cross benchmark correlation study to validate usefulness of high fidelity subtask execution capability
> 4. Table 4, Appendix E also compares online RLVR with offline RL training algorithms.
>
> **Concern: “The claim that the benchmark design is uniquely scalable compared to others that also require manual task specification is overstated.”**
>
> We acknowledge the task authoring bottleneck but maintain that this represents meaningful progress over existing benchmarks requiring both environment infrastructure and task authoring. For any future researchers who would like to use our approach, we would like to mitigate the scalability concern of yours through two arguments.
>
> 1. In case the goal of the researchers is to create more test data, they would most likely require the data to be high in quality and cover all the use cases they are interested in (e.g. on the order of 100 datapoints). In this case, we believe the WARC-Bench approach is perfectly suitable for the level of manual work it involves. We would further argue that, though some manual work is unavoidable, the level of this work required is still significantly less than many other popular computer agent benchmarks with the same level of reality and determinism. For example,
>     - Environment creation in the widely used OS-World[1] benchmark requires writing new mock websites, curating documents and applications, and/or coming up with complex workflows **[takes a few days end to end]**
>     - A similar step of environment creation in WARC-Bench simply involves recording a manual trajectory on a real website **[takes under 10 minutes]**
> 2. Instead, if the goal of the researchers is to generate thousands or tens of thousands of environments, for examples to be used as training environments for RL, we show that even purely synthetic environments can be used for RL training to improve model subtask execution on real-world websites (Section 5). Since synthetic training data can be generated through a much more automated and faster approach, we recommend that method over manual collection for larger scale datasets.
>
> [1] OSWorld: Benchmarking Multimodal Agents for Open-Ended Tasks in Real Computer Environments  https://arxiv.org/abs/2404.07972

---

### Author Response · Authors · 2025-11-25

We thank all the reviewers for their thorough comments.

## Edits made to the paper

1. L092: `We introduce the concept of GUI subtasks - short-horizon tasks that constitute broader browser-based workflows` --> `We formally define GUI subtasks -  short-horizon tasks that constitute broader browser-based workflows` . We did not mean to imply that subtasks are a new task category that we are creating (as evidenced by prior works like MiniWob++), instead we want to formally define the term GUI subtasks for the rest of the paper. This contribution has been edited in the new revision. Our main contribution is to investigate and improve this capability in contemporary models in realistic settings.
3. L097-101: Added `We find that high-fidelity subtask performance on WARC-Bench tracks long-horizon capabilities better than other benchmarks...`
4. Table 2: Added standard deviations to all experimental results on WARC-Bench for dev and test sets.
5. Added examples of subtasks types from WARC-Bench in Appendix A (Subtask Environment and Task Examples).

## Clarifications

1. We have added some samples of what WARC based websites look like [on this link](https://amazing-baklava-78fa98.netlify.app/). WARC archives preserve full client-side interactivity that is sufficient for subtask evaluation – since we are interested in testing client-side UI manipulation skills and not server integration.
2. Results in Table 3 on WebArena are added only to study the generalization of subtask training to long-horizon task execution. We do not expect our models (Ours-7B/72B-RLVR/SFT) to beat models that are trained to perform long-horizon web tasks (like NNet-Nav, AutoWebGLM etc)

## Additional Results

One motivation for this work is to show that GUI sub-agents that specialize in executing subtasks can bring gains to e2e GUI navigation tasks, making subtasks a critical capability to evaluate, diagnose, and improve upon in generalist web agents. We realize that our results in Section 5 might not have sufficiently reflected this, hence we provide additional results for the reviewers to view.

The recommended way for a long-horizon, e2e, web agent to use our model, is to create a hierarchical agent with a **Planner** sub-agent that can break down a long-horizon task into subtasks and redirect them to a performant subtask model (e.g., Ours-72B-RLVR w/ SVA as the executor sub-agent).
|   | Planner Model | Subtask Executor | Agent |  WA-Lite SR% | WA SR% (Table 3) |
|---|---|---|---|:---:|:---:|
| 1 | N/A | Qwen2.5-VL-72B-Instruct | SVA | 16.17 +/- 1.14 | 15.68 +/- 1.08 |
| 2 | Qwen2.5-VL-72B-Instruct | Qwen2.5-VL-72B-Instruct | Hierarchical agent [Planner, Executor] | 22.39 +/- 2.58 | - |
| 3 | Qwen2.5-VL-72B-Instruct | Ours-72B-RLVR | Hierarchical agent [Planner, Executor] | 24.63 +/- 0.75 | - |
| 4 | N/A | Claude-4.0-Sonnet | SVA | 36.32 +/- 1.14 | 37.96 +/- 1.04 |
| 5 | Claude-4.0-Sonnet | Qwen2.5-VL-72B-Instruct | Hierarchical agent [Planner, Executor] | 31.84 +/- 0.87 | - |
| 6 | Claude-4.0-Sonnet | Claude-4.0-Sonnet | Hierarchical agent [Planner, Executor] | 35.32 +/- 1.14 | - |
| 7 | Claude-4.0-Sonnet | Ours-72B-RLVR | Hierarchical agent [Planner, Executor] | 35.82 +/- 1.29 | - |

*Caption: Comparing single-step execution v/s hierarchical planner+executor agent designs. We use WebArena-Lite[1] (WA-Lite) as our evaluation benchmark. This dataset is a subset of 165 WA tasks filtered for evaluator correctness and is comparable in difficulty to the WA benchmark (812 tasks). SVA - Simple agent detailed in Section 3.2. Hierarchical Agent – New hierarchical agent where a planner comes up with a subtask and an executor executes the subtask before returning control to the planner. Map split is excluded in both benchmarks.*

### Conclusions from additional results:

1. For Qwen2.5-VL-72B based agents, we can significantly improve the performance (**+8.46%**) by adopting a hierarchical planner+executor agent design and using a strong executor model (Ours-72B-RLVR) **[Rows 1, 3]**.
2. For strong models like Claude-4.0-Sonnet, further breaking down the task into subtask executions does not bring any gains/downsides in overall performance **[Rows 5,6]**. However, even for Claude-4.0-Sonnet, we can still benefit from a hierarchical design by offloading most of the agent tasks to a cheaper, smaller capable executor model (Ours-72B-RLVR) with no drop in accuracy whilst achieving lesser cost and latency **[Rows 6, 7]**.
3. For long-horizon tasks, the planner model plays a significant role in determining the success rate **[Rows 2, 5 and Rows 3, 7]**. Using a better planner like Claude instead of Qwen2.5-VL-72B, with the same capable executor (Ours-72B-RLVR), significantly improves WA-Lite SR by **+11.19%** points. **[Rows 3, 7]**

We will incorporate these findings in the paper in a subsequent revision.

[1] VisualAgentBench: Towards Large Multimodal Models as Visual Foundation Agents (https://arxiv.org/abs/2408.06327)

---

### Author Response · Authors · 2025-12-03
**Note to ACs (summarizing the rebuttal)**

Dear Area Chair,

We appreciate your time and effort in reviewing our submission on short notice after the OpenReview leak incident. To help with the process, we are providing a summary of the reviews and interactions before the incident.

* All three reviewers liked the presentation of the paper **(scores 3, 3, 4)**.
* **Reviewer igLj (score: 2, contribution ranked as "2: fair")**
  * Concerns:
    * The main concerns of the reviewer were around the soundness, realism and scalability of using WARC files as task environments. They also express concerns about the incrementality of our approach over MiniWob++ and ask for more details around the dataset creation process and data distribution.
  * Response:
    * We have stated in our comments why our approach is sound, and more scalable than other benchmarks. We have provided exact details of our data collection process. We show both qualitative examples and point to empirical results to show why a) MiniWob++ is insufficient to assess subtask / long-horizon capabilities b) WARC files are sufficient to track agent capabilities in realistic settings.
  * Overall, we strongly believe that a lot of reviewer igLj's concerns would be addressed by our comments.

* **Reviewer f6t5 (score: 4, contribution ranked as "3: good")**
  * Concerns:
    * They wanted to see more baselines for models that do well on WebArena (AutoGLM, NNetNav, Go-Browse). They also wanted more details around generalizability of results and training datasets.
  * Response:
    * We note that NNet-Nav and Go-Browse are trained on WebArena environments hence are expected to perform better. We also show that these models do not perform well on our benchmark (WARC-Bench).
    * We provide empirical results to demonstrate the generalizability of our training recipe to OOD environments, and also clarify why the generalization is not expected to work across unseen subtask types.
  * Overall, we believe we provide substantial results to alleviate reviewer f6t5's concerns.

* **Reviewer pxfP (score: 4 → 6, contribution ranked as "2: fair")**
  * Reviewer pxfP was the only reviewer who was able to reply to our comments before the system rollback and they increased our score from 4 → 6.

    *"I appreciate the added experiments and human baseline. Although I am still not fully convinced of the significance of the contribution due to the narrow definition of "subtasks," I like the presentation and the methodology. I have raised the score accordingly."*
  * Concerns:
    * Required more baselines (human, browser-use agents). They wanted to understand how our benchmark compares with other benchmarks like WebVoyager, WebGames etc. They also express that synthetic environments might not replicate realistic environments well enough.
  * Response:
    * We computed human baselines on the development set. We have answered all the reviewer concerns around how our benchmark is different from WebVoyager, WebGames, WebShop. We also explain why and how synthetic environments differ from manually collected environments, yet remain useful for training and evaluation.

---

### Meta-Review · Area_Chair_QNiW · 2026-01-10

**Summary:**

This paper develops a new benchmark for web-navigation tasks. The paper focuses on short-horizon tasks using multi-modal information. It discusses the performance on these tasks using existing LLM/VLM systems and also develops some prototypical models of its own to show that the benchmark is sound and indicative of real-world tasks.

Reviewer iqHj (score 2)
had concerns about (i) the WARC Bench having only static websites, (ii) a predominantly large number of synthetic data, and (iii) the intellectual contributions being minor.

Reviewer f6t5 (score 4)
talked about missing baselines, wanted a discussion on the kinds of websites that were used in this paper and some clarifications on SFT.

Reviewer pxfP (score 4)
wanted comparisons to other short-horizon tasks, human baselines, and some clarifications on dev vs. test set.

These concerns were satisfactorily addressed by the authors in the rebuttal.

**Reviewer Concerns:**

None of the concerns seem outstanding from the rebuttal.

Reviewer iqHj
The rebuttal satisfactorily addressed all concerns. There are plenty of manually selected tasks and the synthetic ones are reliably good. The verification process is sound and quick. The tasks are not exactly static because they mock API calls, javascript etc. when possible. The tasks are more realistic than some existing benchmarks like Miniwob++.

Reviewer f6t5
The rebuttal has provided details on some missing baselines, additional OOD generalization experiments and more details.

Reviewer pxfP
The authors discussed relationships to existing short horizon benchmarks, provided a human baseline and some more details on the experiments/data splits.

**Reviewer Scores:**

Reviewer iqHj
The score would point to an accept.

Reviewer f6t5
The score would point to an accept.

Reviewer pxfP
The score was updated to an accept.

---

### Decision · Program_Chairs · 2026-01-26

Accept (Poster)